# Changes in methodological study characteristics in psychology between 2010-2021

**Ingmar Böschen** *

Department of Research Methods and Statistics, Institute of Psychology, University Hamburg, Hamburg, Germany

* ingmar.boeschen@uni-hamburg.de

## Abstract

In 2015, the Open Science Collaboration repeated a series of 100 psychological experiments. Since a considerable part of these replications could not confirm the original effects and some of them pointed in the opposite direction, psychological research is said to lack reproducibility. Several general criticisms can explain this finding, such as the standardized use of undirected nil-null hypothesis tests, samples being too small and selective, lack of corrections for multiple testing, but also some widespread questionable research practices and incentives to publish positive results only. A selection of 57,909 articles from 12 renowned journals is processed with the *JATSdecoder* software to analyze the extent to which several empirical research practices in psychology have changed over the past 12 years. To identify journal- and time-specific changes, the relative use of statistics based on p-values, the number of reported p-values per paper, the relative use of confidence intervals, directed tests, power analysis, Bayesian procedures, non-standard $\alpha$ levels, correction procedures for multiple testing, and median sample sizes are analyzed for articles published between 2010 and 2015 and after 2015, and in more detail for every included journal and year of publication. In addition, the origin of authorships is analyzed over time. Compared to articles that were published in and before 2015, the median number of reported p-values per article has decreased from 14 to 12, whereas the median proportion of significant p-values per article remained constant at 69%. While reports of effect sizes and confidence intervals have increased, the $\alpha$ level is usually set to the default value of .05. The use of corrections for multiple testing has decreased. Although uncommon in each case (4% in total), directed testing is used less frequently, while Bayesian inference has become more common after 2015. The overall median estimated sample size has increased from 105 to 190.

## Introduction

For decades, many scientists have been voicing well-founded criticism about the general application of statistical methods, specifically for the social sciences. Their critique points to several widespread problems such as the ritualized application of significance tests on nil-null

**Data Availability Statement:** The data and script to reproduce the analysis are available at: dx.doi.org/10.17504/protocols.io.j8nlkk775l5r/v1.

**Funding:** This research was financed by a doctoral grant awarded by the Department of Research Methods and Statistics, Institute of Psychology, University Hamburg, Germany. The funders had no

role in study design, data collection and analysis, decision to publish, or preparation of the manuscript.

**Competing interests:** The author has declared that no competing interests exist.

hypothesis [1–5], the arbitrary selection of mostly small [6, 7] or selective samples [8, 9], the standardized or even wrong interpretation of results [10, 11] or questionable research practices to find the desired (significant) results [12].

An indication of the practical relevance of these concerns in the field of psychology is provided by the large-scaled replication study of the Open Science Collaboration (OSC) [13], which gives empirical evidence for the so-called *Replication Crisis in Psychology*. The OSC conducted 100 high-powered replications of psychological experiments (average power of .92), but only a disappointingly small part lead to the same results as the original studies. The overall replication rates ranged between 36% and 68%, depending on the definition of a replicated result and varied greatly between journals.

The highest replication rates were observed for cognitive studies published by *Psychological Science*. Fifty-three per cent of the replicated experiments from this journal lead to effects of the original direction and p-values < .05. Also, 53% were subjectively considered a successful replication and 92% reached meta analytical p-values < .05. One qualification about this result is the possibility that the original studies have inflated effect sizes due to publication, selection, reporting, or other biases [13].

The widespread problematic or even flawed application of statistical inference and conclusions made therewith, led Ioannidis to the general, pessimistic conclusion that *"most published research findings are false"* [14]. In the light of the OSC study, however, this statement appears to be exaggerated for psychology, but it points to severe credibility problems.

Journals have developed some general and individual quality standards intended to increase the reliability of their publications. For quite some time, peer review has been an important standard in quality assurance and is undoubtedly irreplaceable for a credible science. However, the replication crisis was not prevented by the peer-review process. Other journal requirements designed to ensure research quality vary from journal to journal and have and will continue to evolve over time. These requirements mostly relate to justification of sample size, precision of description of data preparation, reporting of statistical test results, reporting of effect sizes and confidence intervals, publication of evaluation protocols and data, preregistration of research projects. Taken together, the standards set by journals have a strong influence on how scientists plan and conduct their studies and how they report and interpret their findings. However, in many cases, even these so-called 'best practices' have apparently not been sufficient to ensure the replicability of a study result in the past.

## What is going wrong? Possible causes of the replication crisis

Despite many great scientific achievements, something seems to be going wrong in some areas of scientific reasoning. The here listed critique is not new, but still of major relevance for psychological research practice, as the low replication rates in the OSC study suggest. One of the most important issues is certainly the overly ritualized use of undirected nil-null hypothesis significance testing.

Most frequentist methods have in common that they are based on assumptions about the sampling process and the distributional properties of involved variables. Based on these assumptions and a given null hypotheses ($H_0$), the theoretical distribution of an estimator $\hat{\theta}$ and its variance (inaccuracy) $\hat{\sigma}_{\hat{\theta}}^2$ is determined. The realized estimator $\hat{\theta}$ can then be checked for random deviation from the theoretical value $\theta_0$, which was specified in $H_0$. It should be emphasized that, unlike in practice, the empirical effect estimator $\hat{\theta}$ can be tested against any value of $\theta_0$. $\theta_0$ can be the difference in means of two groups, a correlation, model fit index, or any other statistical measure of interest.

Predefined critical values of the determined test statistic serve as a threshold for a binary decision about $\theta_0$. If the test statistic falls within the rejection region, the $H_0$ is rejected. Otherwise $H_0$ is not rejected, which is not to be confused with proved or accepted.

## Low power = small samples = high uncertainty

Undoubtedly, the replication crisis is a symptom of a situation of great uncertainty that can be caused by low statistical power, noisy measurements and/or questionable research practices. An obvious way to reduce uncertainty in scientific contexts is to increase the measurement accuracy of the estimated parameters $\theta$. The test strength or power ($1 - \beta$ error) of a statistical test is the theoretical probability of correctly rejecting the null hypothesis—getting a significant result—with an a priori set $\alpha$ error and planned or realized sample size $N$, assuming an effect of a certain size $\delta_{true} = \theta_{true} - \theta_0$. Conversely, this idea is suitable for estimating the required minimum sample size in order to discover effects (deviations from the $H_0$) of a certain minimum size $\delta_0$, or to estimate the optimal sample size, to achieve a power or length of a $1 - \alpha$ confidence interval, with an adequate $\alpha$ level.

A power analysis for a two-sided comparison test of means (T-test) with the R function *power.t.test()* yields an optimal size of $n_{opt} > 99.08$ units per group to identify a medium-sized effect of $\delta = .4$ standard deviations, with an $\alpha$ level of .05 and a power of .8. Studies that focus on smaller effects or work with lower $\alpha$ levels require much larger samples in order to identify the effects (get a significant result) with the same power.

In 1962, Cohen found that most of the studies published in the *Journal of Abnormal and Social Psychology* (46%) had low power ($< .5$). When one posits medium effects in the population (Cohen used $\delta = .5$ sd) the studies have, on average, a slightly less than 50-50 chance of successfully rejecting their major null hypotheses [6]. None of the investigated studies had a power $> .5$, when considering small effects ($\delta = .25$ sd).

Again, 24 years later Sedlmeier & Gigerenzer [7] examined the power of psychological studies of several journals and found that the median power had decreased even further. Among other critics, Gelman & Carlin [15] show that low-powered studies have fundamental problems and introduce the terms type M (magnitude) and type S (sign) error. *"When researchers use small samples and noisy measurements to study small effects—as they often do in psychology as well as other disciplines—a significant result is often surprisingly likely to be in the wrong direction and to greatly overestimate an effect"* [15]. The practical relevance of these error types is supported by the OSC report, in which the replication effects were half the size of the original effects and some effects pointed in the opposite direction of the original effects.

One approach to sample size planning is to estimate the desired (in-) accuracy of the parameter to be estimated and use variance estimates, for example from previous studies or standard values. A researcher might want to estimate the effect of a phenomenon with a 95% confidence interval that has an accuracy of ± 3.5 units (points, milliseconds, etc.).

Further, focusing on the $\beta$ error (not rejecting $H_0$, although it is false) or test power enables the determination of a necessary or even optimal sample size to identify the effect sizes of interest. This approach also permits an interpretation of insignificant results as an indication (not as evidence) for the absence of an effect of a certain size, if the power of the test is high.

## The undirected nil-null test scenario

A common misconception about the null hypothesis test in psychology and the social sciences is that the null hypothesis is to be equated with a zero effect ($\theta_0 = 0$), or zero correlation [1, 10, 16]. This type of hypothesis represents a special case and is also known as the nil-null

hypothesis, the nil-hypothesis [3] or the point null hypothesis [2]. The term nil-null refers to a test setting that hypothesizes a true null effect or zero correlation.

Since two or more groups of units are never completely alike and most statistical estimates $\hat{\theta}$ are weakly consistent estimates, meaning that their variance/inaccuracy $\sigma_\theta^2$ converges to 0 when the number of observations (N) grows to infinity ($\lim_{n\to\infty} \sigma_{\hat{\theta}} = 0$), Cohen points out that very large samples always lead to the rejection of any precise point-zero hypotheses [17]. Twenty-eight years after his first study on the power of psychological studies [6], Cohen reiterated his recommendation to psychologists, that they should include thoughts about the direction and strength of the effect to be discovered, when planning a study [17]. *"The null hypothesis can only be true in the bowels of a computer processor running a Monte Carlo study (and even then a stray electron may make it false)"* [17].

This position coincides with Meehl's paradox [1] of the opposing application scenarios of the null hypothesis significance test and the handling of uncertainty, by psychologists and physicists. *"While physicists generate a specific prediction about their model and test whether the measurement results deviate significantly from the predictions, psychologists usually test their data for a zero effect and this in addition mostly in an undirected manner in order to underpin their model with significant results. Further, it had been more explicitly recognized that what is of theoretical interest is not the mere presence of a difference (i.e., that $H_0$ is false) but rather the presence of a difference in a certain direction"* [1].

Psychologists seem to be sceptical about generating a priori set theoretical point estimates or borders of interest apart from zero. This surely limits them to use uninformed nil-null hypothesis tests and seek for a significant result, which is then assessed as support for their theory.

It is common practice to report the p-value of a statistical test result. The p-value is the probability of obtaining a value $\hat{\theta}$ or more extreme deviations from $\theta_0$, when $H_0$ is in fact true. The p-value is compared with a previously defined $\alpha$ level (the probability of incorrectly rejecting $H_0$) to make a binary decision. If a p-value is smaller than the $\alpha$ level, the test result is reported as significant, if $p > \alpha$ the $H_0$ is not rejected. In practice the $\alpha$ level is mostly set to the undefined standard of .05.

Another frequentist approach was developed by Neyman [18] three decades before Meehl's criticism. Neyman introduced the $1 - \alpha$ confidence interval for a parameter $\theta$. Cox & Hinkley [19] describe confidence intervals in terms of a random sample. *"Were this procedure to be repeated on numerous samples, the fraction of calculated confidence intervals (which would differ for each sample) that encompass the true population parameter would tend toward $1 - \alpha\%$"*. Besides the additional information about uncertainty, it includes a test on a statistical null hypothesis. If the $1 - \alpha$ confidence interval contains the value of $\theta_0$, the test statistic leads to a p-value $> \alpha$. If $\theta_0$ is not included, the resulting p-value is below $\alpha$, what is called a significant deviation from $H_0$. In fact this simple check does not add any information to the p-value based procedure. Rather, the empirical effect and the width of the interval should be in focus.

In 1996, Schmidt [20] calls for the replacement of significance tests by point estimates and confidence intervals, as this approach leads to much more informative results. Although confidence intervals have been around for a long time, in 2014 Cumming [21] still encourages an increased use of effect estimates combined with confidence intervals and a focus on their width, which he jovially called 'The New Statistics'.

A rather seldom proclaimed procedure are equivalence tests. Equivalence tests enable to falsify predictions about the presence, and declare the absence, of meaningful effects [22]. This type of testing differs from the standard method in that interval hypotheses are tested and support for the null hypothesis can be generated. In principle, equivalence tests demand an a

priori set definition of trivially small, meaningless effects. An undirected hypothesis is then tested with two one-sided tests against the bounds of the predefined interval of meaningless effects. Non-inferiority and superiority trials are another type of interval null hypotheses primarily used in medical treatment evaluation. The null hypothesis in non-inferiority trials states that a new treatment is worse than an old by more than $-\Delta$, where $-\Delta$ is the 'non-inferiority margin' [23].

Although it is the most commonly applied threshold, several authors have expressed their doubts about the usefulness of a fixed alpha error of .05 for the discovery of new effects and have proposed to lower it to .005 [24–26]. These and several other authors [21, 27–29] have even proposed to move away from frequentist methods entirely and to use Bayesian statistics instead (e.g. Bayes factor, posterior distribution). Compared to standard null hypothesis testing, Bayesian methods update prior assumptions about an effect or the probability of a hypothesis with the incoming data. This approach allows more intuitive statements about the probability of a hypothesis given the data, or the ratio of the probabilities of competing hypotheses given the data. For a detailed comparison of the frequentist and Bayesian approach, the reader is referred to the literature.

As a consequence of the ongoing debate about the utility of p-values, the editors of the journal *Basic and Applied Social Psychology* have banned p-values and hypothesis tests in 2015 [30]. However, according to Lakens [31], this change has not been accompanied by a shift toward Bayesian methods nor confidence intervals and has tended to degrade the quality of studies published in the journal as measurement inaccuracy is vanished.

## Questionable research practices

Questionable research practices (QRPs) that manipulate the data or the analytical process, until the desired effects are found, are widespread [12, 32]. After listing several potential ways to manipulate the data collection and analytical processes, Simmons et al. conclude that in many cases it is more likely that a researcher will find false evidence for the presence of an effect than that he/she will find evidence that there is no effect [12].

For example, optional stopping refers to a practice that is applied if a result is not significant. Additional cases are drawn until the desired result has been achieved, without reporting or correcting for this sampling procedure (e.g. sequential testing). Further, unplanned subgroup comparisons can easily be carried out, especially in large data sets with many potential dependent variables, and only the significant effects are reported later. HARKing (Hypothesizing After Results are Known) is a special form of this approach. HARKing refers to the report of a post hoc hypothesis, as if it were actually an a priori hypothesis [33]. Further, post hoc exclusion and inclusion criteria can be defined until the results meet one or many specific requirements. The file drawer problem, which was introduced by Rosenthal [34], refers to the fact that mostly studies that 'worked' are brought to publication and that there is therefore a bias towards 'positive' results in the literature.

In 2012, John et al. surveyed over 2,000 American psychologists and found that the percentage of respondents who reported to have engaged in QRPs was surprisingly high [35]. For certain practices, their inferred actual estimates approached 100%, which suggests that these practices may constitute the de facto scientific norm [35]. It should be noted, that the survey did not ask how often these practices were used, only whether they were ever used at all. Ten years later, Fox et al. estimated that 18% of American psychologists have used at least one QRP in the past 12 month and that QRP users are a stigmatized sub-population of psychologists [36].

An effective solution to minimize the use of HARKing and optional stopping is to determine the research questions and analysis plan before observing the research results—a process called preregistration [37]. However, it remains questionable whether preregistration also counteracts the file drawer problem, since authors and editors can continue to make the publication dependent on the result. In registered reports a study protocol containing the hypotheses, planned methods, and analysis pipeline undergoes peer review before the data collection and the results are published regardless of whether the hypotheses are supported or not [38]. By analyzing the first hypothesis of standard and registered reports, Scheel et al. show that the latter publication process leads to significantly lower success rates (96% positive results vs. 44% positive results in registered reports) and therefore reduces publication bias and/or $\alpha$ error inflation [38].

Steegen [39] proposed multiverse analyzes as an approach to coping with the researcher's degrees of freedom within an analytical process. A multiverse analysis displays the stability or robustness of a finding, not only across different options for exclusion criteria, but across different options for all steps in data processing [39].

## Multiple testing

Since the probability of receiving at least one false positive (significant) test result increases with each statistical test performed on the same data ($\alpha$ error accumulation), the $\alpha$ level or p-values should usually be adjusted/corrected for multiple tests of the same family (family-wise error rate). Thus, in the absence of strong a priori expectations about the tests that are relevant, this alpha inflation can be substantial and be a cause for concern [40]. It is rarely emphasized that this also applies to the reporting of multiple 1-$\alpha$ confidence intervals. A simple, rigid/conservative method for an $\alpha$ level/p-value adjustment is the Bonferroni correction, in which the underlying $\alpha$ level/empirical p-values are corrected with the total number of tests carried out. There are various other, more powerful methods for adjusting the $\alpha$ level/p-values for multiple testing, such as the sequential method by Holm [41] or Benjamini & Hochberg [42]. If a large number of statistical tests is carried out on the same data, with constant effects and corrected $\alpha$ levels, then the sample sizes must be increased in each case in order to discover the effects with the same power. Another technique to control the $\alpha$ error in multi group comparisons is partial pooling, which is applied in multilevel models with undiminished power [43].

## Selective samples

Although sample characteristics are rather unrelated to the low replication rates of the OSC report, sampling may be a critical issue in drawing valid conclusions about humanity. Henrich et al. point out that psychological research is mostly based on samples from and written by authors from western educated industrial rich and democratic (WEIRD) societies [9]. Their analysis of articles by the top journals in six sub-disciplines of psychology from 2003 to 2007 shows that although WEIRD residents represent only a small proportion of the world population (12%), 73% of first authors were at American universities, and 99% were at universities in Western countries [9]. Annett examined studies published in the *Journal of Personality and Social Psychology* in 2007 and found that in 67% of U.S. and 80% of non-U.S. studies, the samples consisted of undergraduate psychology students [8], which is certainly an even more specific kind of humanity.

## Method

The extent to which some of the methodological features highlighted here have changed over the past 12 years is examined using a collection of 57,909 psychological research articles from 12 journals.

The R [44] package *JATSdecoder* [45] is used to extract the methodological study features in focus here. *JATSdecoder* is a general toolbox which facilitates text extraction and analytical tasks on NISO-JATS coded XML documents [46]. *JATSdecoder* consists of two main modules. The *JATSdecoder()* function extracts metadata such as the title, abstract, author, keywords, country of origin, reference list and the main text, which is stored as a vector of sections. The *study.character()* function mainly processes the text of the methods and results sections and extracts important study characteristics using expert-guided heuristics. It has been demonstrated that *study.character()* reliably extracts statistical results reported in text [47] and methodological study features such as the statistical methods applied, the $\alpha$ level and correction procedures for multiple testing, a priori and a posteriori power and the test sidedness [48]. For example, the statistical methods extraction algorithm uses an n-gram bag of words approach for a set of pre-specified inclusion terms (e.g., test, interval, bayes, analysis). This allows for accurate identification of most statistical procedures while keeping the result space open. The underlying sample size is estimated based on reports within the abstract (textual and numerical representations) and by a user-definable quantile of all extracted degrees of freedom, that are detected within the reported statistical test results within the text. The .9-quantile is used here to reduce overestimation by reports that mostly contain subgroup analyses or mishandling of repeated measures as independent units. It should be emphasized that the latter function has not yet been evaluated and is still in the developmental stage.

All research articles published between 2010 and 2022 by ten highly renown journals (*Behavioral Neuroscience*, *Depression & Anxiety*, *J. Abnormal Psychology*, *J. Child Psychology*, *J. Family Psychology*, *J. Management*, *Personality and Social Psychology Bulletin*, *Psychological Medicine*, *Psychology & Aging*, *Psychophysiology*) of five psychological subdisciplines (Biological, Clinical, Developmental, Social, Work and Organizational Psychology) were downloaded manually with the license of the University of Hamburg as a PDF version. The Content ExtRactor and MINEr *CERMINE* [49] was then used to convert the original PDF files into XML files, structured in the *Journal Article Tag Suite* standard markup (NISO-JATS) [50]. In order to further include articles published in open access journals in the analysis, research articles from *Frontiers in Psychology* and *Plos One* were selected. The full PubMed Central (PMC) database was downloaded in bulk from https://ftp.ncbi.nlm.nih.gov/pub/pmc/oa_bulk/ in XML format in January 2022 (4,048,608 files), processed with *JATSdecoder* [45], and reduced to articles that were published by *PLoS One* and *Frontiers in Psychology* between 2010 and 2021 and type tagged as 'research-article'. Articles on psychology from *Plos One* were selected using a search task with the pattern 'psycholog' on the lowerized title, abstract, keyword, subject and affiliation tags.

Articles that contain any of the search terms 'meta-analysis', 'review', 'letter to the editor', 'corrigendum' or 'commentary' in their title and that do not contain any statistical result in the main text are removed in order to select only empirical research articles for the analysis.

Global and journal-specific trends and changes in study characteristics are analyzed over time with conditional frequency tables and bar plots. The distribution of the number of reported p-values per journal is shown in a box plot.

In terms of reporting style, a distinction is made between the results as follows. While extractable results with p-value contain a p-value pointing to a number with an operator (e.g.: $p < .05$), computable p-values can also be extracted from incompletely reported results (e.g.: $t$

(32) = 1.96). A checkable result must be computable and also contain a p-value (e.g.: $t(32) = 1.96$, $p = .05$). It must be emphasized that only results reported in the text can be extracted. Results in tables and figures are not part of the analysis.

To analyze the use of standard effect measures (Cohen's, $\eta^2$, beta coefficients), the corpus of studies is limited in each case to those that cite appropriate procedures (t-test, ANOVA, regression).

Because *JATSdecoder*'s extraction heuristic of the $\alpha$ level cannot distinguish between nominal and corrected values, the extracted maximum value is analyzed in articles that contain extractable p-values. For articles that use multiple $\alpha$ levels, only the maximum value is extracted for analysis.

Since the the Bayesian information criterion (BIC) is used for model selection and not for statistical inference, it is removed from the extracted statistical methods before analyzing changes in the use of Bayesian inference methods.

In an evaluation study [48] the accuracy of all applied extraction heuristics for methodological study characteristics was high (from.85 for $\alpha$ levels to .993 for directed testing).

Global changes in study characteristics of content published before and after 2015 are reported with 99.9% confidence intervals for differences in proportions and 99.9% bootstrap confidence intervals for differences in medians (20,000 resamples). Journal-specific analyses are not reported with p-values nor confidence intervals because the respective samples represent a complete and not a random sample of articles from the respective journal.

To facilitate the analysis of changes in the country of origin the *countrycode* [51] package is used to convert country names to continents, which are then transformed to WEIRD and non-WEIRD involved countries of origin (except for Israel and New Zealand, which are manually set to WEIRD). Multicore processing is performed using the R package *future.apply* [52].

## Results

The final article collection comprises 57,909 articles. 5,900 out of 63,809 initial articles were removed, as they contain one of the exclusion search terms within the title or do not contain any extractable statistical result. The journal-based distribution of yearly released research articles is shown in Fig 1. Both full open access journals (*Frontiers in Psychology* and *PLoS ONE*) published 77% of all included articles. Along with a steady increase of yearly published articles by *Frontiers in Psychology* and *PLoS One*, *PLoS One* shows to have a slump in psychological research content in 2015. The closed access journals appear to have a more constant number of released research articles per year.

The article set is divided into two 6-year publication intervals to analyze global changes in study characteristics of psychological research. Table 1 shows the relative use of the here targeted study characteristics for the selected articles that were released before and after 2015, as well as for the full article collection.

Overall, the proportion of articles that report p-values has decreased from 92% in and before 2015 to 82% in publications after 2015. The proportion of articles that report multiple studies has decreased from 17% to 13%. The median number of reported p-values per study in articles with p-values has decreased from 14 to 12. Compared to publications in and before 2015, fewer articles contain results that are reported in a manner, that enables a recomputation (69% vs. 58%) and a consistency check of p-values (67% vs. 55%). Overall, the median proportion of reported p-values that are below.05 is constantly high (69%) and even higher (74%) when only computable p-values are considered. That is, 50% of the articles contain more than these proportions of significant results at $\alpha = .05$. Hardly any use of $\alpha$ levels below.05 is observed.

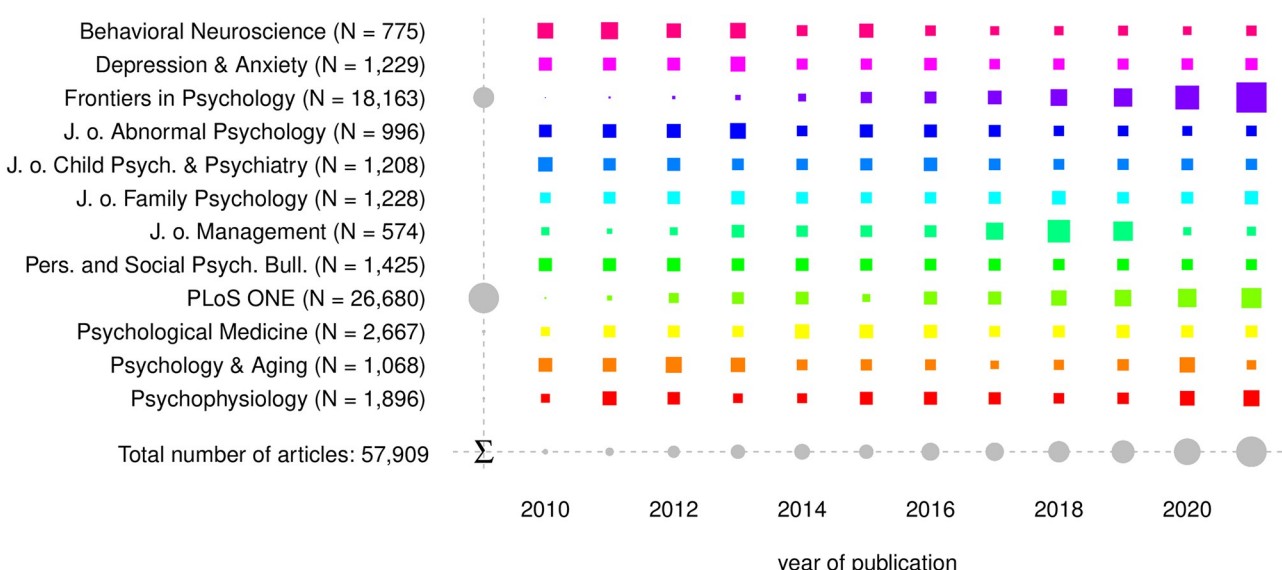

**Fig 1. Number of yearly released research articles by journal and year.**

Within all included articles, the reporting of confidence intervals has increased from 21% to 32%. Reports of power analysis and the application of Bayesian inference have more than doubled, from 5% to 11% and from 2% to 5%, respectively. Corrections for multiple testing are reported in about one quarter of all articles, with a slight decrease from 27% to 23%, although multiple testing is applied in almost every article. Directed testing is rarely reported, with a

**Table 1. Change in study characteristics in research articles with statistical results before and after 2015.**

| Feature | ≤ 2015 | > 2015 | total | Δ | 99.9% CI for Δ |
|---|---|---|---|---|---|
| N articles in initial selection | 21,238 | 42,571 | 63,809 | 21,333 | |
| N empirical research articles | 19,443 | 38,466 | 57,909 | 19,023 | |
| proportion of articles with with p-value/s | .92 | .82 | .85 | -.098 | [-.108; -.088] |
| → proportion of articles with multiple studies | .17 | .13 | .15 | -.047 | [-.058; -.036] |
| → median number of p-values per study | 14 | 12 | 13 | -2 | [-2; -1] |
| → proportion of articles with recomputable p-value | .69 | .58 | .62 | -.11 | [-.125; -.095] |
| → proportion of articles with checkable p-value | .67 | .55 | .60 | -.12 | [-.135; -.105] |
| → median proportion of reported $p < .05$ | .69 | .69 | .69 | -.006 | [-.018; .007] |
| → median proportion of computable $p < .05$ | .73 | .75 | .74 | -.011 | [-.026; .018] |
| → proportion of articles with alpha level < .05 | .03 | .02 | .02 | -.006 | [-.011; -.002] |
| → proportion of articles with alpha level < .01 | .01 | .01 | .01 | -.004 | [-.007; -.001] |
| proportion of articles with confidence interval | .21 | .32 | .28 | .106 | [.092; .119] |
| proportion of articles with power analysis/value | .05 | .11 | .09 | .062 | [.054; .07] |
| proportion of articles with Bayesian analysis | .02 | .05 | .04 | .023 | [.017; .028] |
| proportion of articles with correction for multiple testing | .27 | .23 | .24 | -.045 | [-.058; -.031] |
| proportion of articles with one sided test | .05 | .03 | .04 | -.019 | [-.025; -.013] |
| proportion of articles with extractable sample size | .83 | .77 | .79 | -.063 | [-.075; -.05] |
| → median of extracted sample sizes | 105 | 190 | 151 | 85 | [74; 95] |

Note: CIs for Δ represent confidence intervals for differences in proportions and bootstrap confidence intervals for differences in medians based on 20,000 resamples.

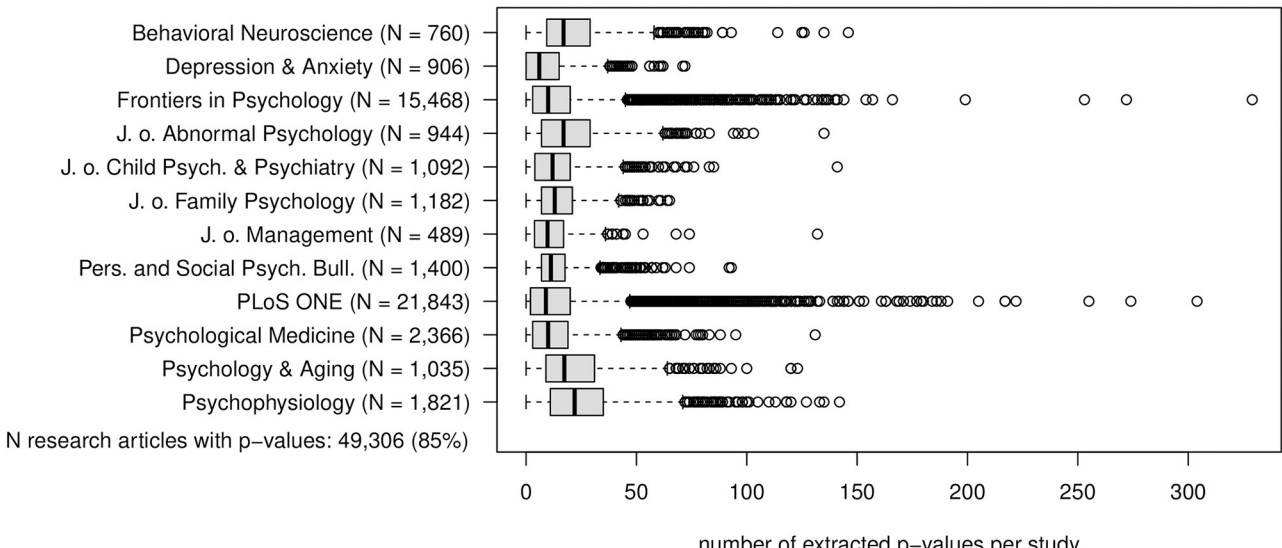

**Fig 2. Number of articles with p-values and distribution of number of extracted p-values per study by journal.**

decrease from 5% to 3%. In 79% of all articles, a sample size is estimated by *JATSdecoder*. Articles published after 2015 show to have much higher median sample sizes (190) than articles published in and before 2015 (105).

More detailed insight per journal and year is given for every feature in the following sections.

## The reporting of p-values

Extractable p-values are detected in 49,306 (85%) articles. The full distribution of the number of extracted p-values per study is displayed for every journal in Fig 2. It is evident that many studies report so many p-values that a correction for multiple testing is required to control the false positive rate. Every here included journal contains several articles with more than 50 p-values per study. Some studies in articles published by *Frontiers in Psychology* and *PLoS One* even contain more than 200 p-values.

A more precise insight to the amount and properties of reported p-values is displayed for every journal in Table 2. It contains the proportion of articles that report p-values within text, the median and total number of extractable, computable and checkable p-values per journal, as well as the proportion of computable p-values that fall below .05, .01 and .001.

Overall, 85% of all analyzed articles report at least one p-value. The lowest rate is observed for *Depression & Anxiety* (74%). The median number of p-values per study within articles that report p-values ranges between 10 in *Depression and Anxiety* and 22 in *Psychophysiology*. The median number of checkable p-values per study is lower for every journal. Still, 50% of all articles that contain p-values report more than 8 computable and checkable p-values. In total, 67% of all computable p-values (two-sided) are smaller than .05, 49% are below .01 and 35% below .001.

## The reporting of effect sizes

Table 3 displays the relative frequencies of reported standard effect measures (Cohen's d, $\eta^2$, $\hat{\beta}$/OR) within articles using the corresponding statistical procedure (t-test, ANOVA,

**Table 2. Journal specific properties of extracted raw, computable and checkable p-values within articles that contain any statistical result.**

| Journal | N articles | % with p-value | median number of p-values per study | | | total number of p-values | | | proportion of computable p-values | | |
|---|---|---|---|---|---|---|---|---|---|---|---|
| | | | p-value | comp. | check. | p-value | comp. | check. | p<.05 | p<.01 | p<.001 |
| Behavioral Neuroscience | 775 | 98% | 17 | 12.8 | 12 | 22,427 | 15,199 | 14,336 | 63% | 43% | 29% |
| Depression & Anxiety | 1,229 | 74% | 10 | 6 | 6 | 12,005 | 4,980 | 3,827 | 65% | 45% | 32% |
| Frontiers in Psychology | 18,163 | 85% | 12.5 | 8 | 8 | 303,221 | 163,286 | 150,908 | 67% | 50% | 37% |
| J. Abnormal Psychology | 996 | 95% | 18 | 10 | 10 | 21,108 | 10,081 | 9,447 | 66% | 47% | 33% |
| J. Child Psych. & Psychiatry | 1,208 | 90% | 13 | 7 | 7 | 17,387 | 7,485 | 6,906 | 67% | 45% | 31% |
| J. Family Psychology | 1,228 | 96% | 13.5 | 6 | 5 | 19,326 | 7,371 | 6,643 | 64% | 45% | 30% |
| J. Management | 574 | 85% | 11 | 4 | 3.5 | 8,032 | 2,349 | 1,779 | 73% | 61% | 51% |
| Pers. and Social Psych. Bull. | 1,425 | 98% | 11.5 | 7.7 | 7.2 | 48,804 | 32,877 | 31,057 | 69% | 48% | 35% |
| PLoS ONE | 26,680 | 82% | 12 | 7.5 | 7 | 409,800 | 146,175 | 133,748 | 67% | 49% | 35% |
| Psychological Medicine | 2,667 | 89% | 11 | 6 | 6 | 35,153 | 11,185 | 10,125 | 65% | 45% | 32% |
| Psychology & Aging | 1,068 | 97% | 18 | 13 | 11.9 | 27,818 | 18,067 | 16,432 | 72% | 54% | 40% |
| Psychophysiology | 1,896 | 96% | 22 | 16 | 15 | 51,121 | 34,240 | 32,226 | 70% | 50% | 35% |
| Total | 57,909 | 85% | 13 | 8 | 8 | 976,202 | 453,295 | 417,434 | 68% | 49% | 35% |

regression) and their increase factors for articles published before and after 2016. The proportions of articles reporting a standard effect measure are calculated based on the number of articles in which *JATSdecoder* identified an appropiate statistical method (t-test, ANOVA, Regression).

The reporting of effect sizes has increased for all measures involved and differs greatly between journals. All journals show inceased reporting rates for Cohen's d in articles that report the use of t-tests. Wheras 4% of the articles by *Behavioural Neuroscience* that report t-test results and that where published before 2016 report Cohen's d, 50% of articles published by *Personality and Social Psychology* after 2015 report it within the text corpus. Global textual reporting rates for $\eta^2$ in articles with ANOVA and $\hat{\beta}$/OR in articles with regression analysis have also mostly increased. Each of the effect measures presented here is most commonly

**Table 3. Proportion of articles that report standard effect sizes in studies with t-test, ANOVA and regression analysis.**

| Journal | Cohen's d in t-test | | | $\eta^2$ in ANOVA | | | $\hat{\beta}$/OR in regression | | |
|---|---|---|---|---|---|---|---|---|---|
| | ≤ 2015 | ≥ 2015 | factor | ≤ 2015 | ≥ 2015 | factor | ≤ 2015 | ≥ 2015 | factor |
| Behavioral Neuroscience | 0.04 | 0.13 | 3.09 | 0.04 | 0.18 | 4.72 | 0.09 | 0.11 | 1.20 |
| Depression & Anxiety | 0.21 | 0.24 | 1.18 | 0.16 | 0.12 | 0.74 | 0.25 | 0.23 | 0.92 |
| Frontiers in Psychology | 0.21 | 0.30 | 1.43 | 0.49 | 0.55 | 1.14 | 0.37 | 0.47 | 1.28 |
| J. Abnormal Psychology | 0.30 | 0.50 | 1.65 | 0.23 | 0.40 | 1.69 | 0.21 | 0.28 | 1.29 |
| J. Child Psych. & Psychiatry | 0.22 | 0.30 | 1.36 | 0.29 | 0.28 | 0.94 | 0.37 | 0.40 | 1.08 |
| J. Family Psychology | 0.16 | 0.19 | 1.18 | 0.08 | 0.16 | 1.92 | 0.30 | 0.43 | 1.45 |
| J. Management | 0.08 | 0.15 | 1.95 | 0.08 | 0.16 | 2.12 | 0.48 | 0.65 | 1.36 |
| Pers. and Social Psych. Bull. | 0.40 | 0.57 | 1.45 | 0.57 | 0.73 | 1.28 | 0.84 | 0.84 | 0.99 |
| PLoS ONE | 0.12 | 0.16 | 1.42 | 0.26 | 0.31 | 1.19 | 0.25 | 0.22 | 0.91 |
| Psychological Medicine | 0.15 | 0.21 | 1.41 | 0.12 | 0.17 | 1.41 | 0.25 | 0.29 | 1.16 |
| Psychology & Aging | 0.26 | 0.39 | 1.50 | 0.39 | 0.58 | 1.49 | 0.24 | 0.37 | 1.50 |
| Psychophysiology | 0.16 | 0.34 | 2.11 | 0.37 | 0.41 | 1.09 | 0.34 | 0.44 | 1.29 |
| Total | 0.15 | 0.24 | 1.54 | 0.32 | 0.42 | 1.33 | 0.40 | 0.44 | 1.09 |

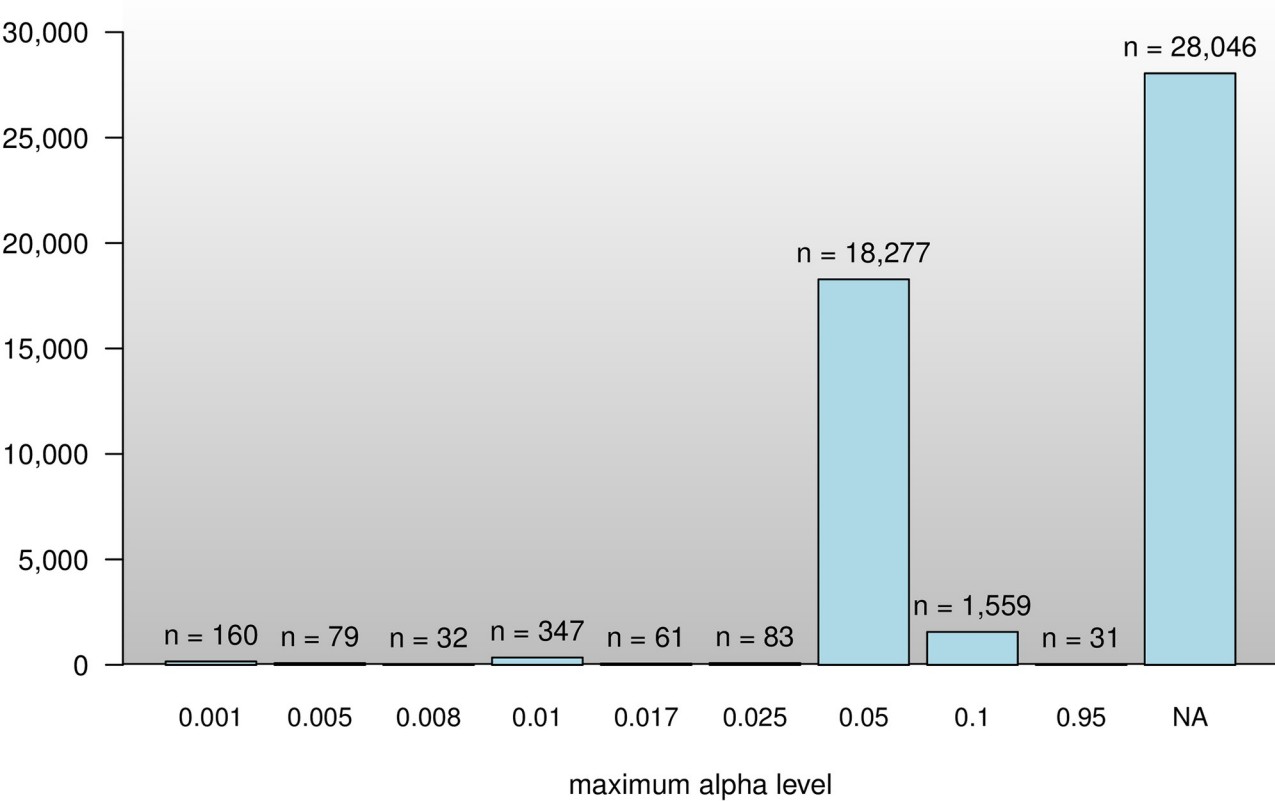

**Fig 3. Distribution of maximum $\alpha$ levels extracted over 30 times.**

reported in *Personality and Social Psychology* with 84% of studies using an regression also reporting $\hat{\beta}$ or OR within the text corpus.

### $\alpha$ level

Most of the articles (57%) that contain p-values do not contain an extractable report of an $\alpha$ level but mostly make implicit use of the standard $\alpha$ level level of .05. In order to rule out incorrect and mostly corrected $\alpha$ levels, Fig 3 shows the distribution of the extracted maximum $\alpha$ levels that were extracted over 30 times. $\alpha$ levels other than.05 are extremely rare. The second most common detected maximum $\alpha$ level is 10%. Standard $\alpha$ levels of .01 and .001 are hardly detected in any article (N = 347 and N = 160). Most of the extracted values of .005 are corrected $\alpha$ levels in studies that analyze voxels in fMRI analyzes with a nominal $\alpha$ of .05. Some articles (N = 31) report using a 95% significance level, but actually use a.05 $\alpha$ level.

Table 4 shows the relative frequency of articles with p-values that report the use of a maximum $\alpha$ level below.05. There do not seem to be any notable changes in the use of the $\alpha$ level over time. The highest proportion of articles with p-values below.05 is detected in the articles of the *Journal of Management* in 2014 (7%). However, it remains unclear whether this is due to corrected or nominal levels of $\alpha$.

**Table 4. Relative frequency of extracted maximum $\alpha$ levels $<$ .05 in articles with p-values.**

| Journal | 2010 | 2011 | 2012 | 2013 | 2014 | 2015 | 2016 | 2017 | 2018 | 2019 | 2020 | 2021 | Sum |
|---|---|---|---|---|---|---|---|---|---|---|---|---|---|
| Behavioral Neuroscience | .04 | .02 | .00 | .00 | .00 | .04 | .00 | .04 | .00 | .02 | .00 | .02 | .02 |
| Depression & Anxiety | .00 | .00 | .04 | .04 | .06 | .02 | .01 | .00 | .00 | .02 | .05 | .03 | .03 |
| Frontiers in Psychology | .01 | .03 | .02 | .03 | .03 | .02 | .03 | .02 | .03 | .02 | .02 | .02 | .02 |
| J. Abnormal Psychology | .04 | .02 | .03 | .02 | .06 | .02 | .00 | .01 | .03 | .01 | .00 | .04 | .02 |
| J. Child Psych. & Psychiatry | .03 | .01 | .01 | .02 | .00 | .04 | .01 | .00 | .02 | .02 | .00 | .03 | .02 |
| J. Family Psychology | .00 | .00 | .01 | .03 | .00 | .01 | .00 | .00 | .02 | .00 | .00 | .00 | .01 |
| J. Management | .00 | .00 | .04 | .00 | .07 | .03 | .00 | .00 | .00 | .00 | .00 | .03 | .01 |
| Pers. and Social Psych. Bull. | .01 | .02 | .01 | .01 | .02 | .01 | .00 | .01 | .00 | .01 | .00 | .01 | .01 |
| PLoS ONE | .02 | .03 | .04 | .03 | .03 | .03 | .03 | .02 | .02 | .02 | .01 | .01 | .02 |
| Psychological Medicine | .03 | .02 | .01 | .03 | .04 | .02 | .01 | .01 | .03 | .02 | .00 | .02 | .02 |
| Psychology & Aging | .00 | .02 | .02 | .01 | .01 | .01 | .01 | .07 | .03 | .01 | .00 | .04 | .02 |
| Psychophysiology | .03 | .00 | .01 | .01 | .01 | .01 | .01 | .03 | .02 | .03 | .02 | .01 | .01 |
| Total | .02 | .02 | .03 | .02 | .03 | .02 | .02 | .02 | .02 | .02 | .02 | .02 | .02 |

## Use of confidence intervals

Within articles that contain any extractable result, the proportion of articles that report confidence intervals has increased (from 21% before 2016 to 32% after 2016) with a peak of 35% in 2021. All journals report more confidence intervals today than they did before 2016. Table 5 shows the relative frequency of detected reports of confidence intervals in articles with p-values. The highest rate of use is observed in *Personality and Social Psychology Bulletin* (90%) which seems to have changed reporting standards in 2015, when reporting rates increased strongly. Between 2010 and 2021, the relative use of confidence intervals increased the most in the *Personality and Social Psychology Bulletin* (from 19% to 90% and the *Journal of Abnormal Psychology* (from 22% to 65%). The lowest rate of articles with confidence intervals is found in *Behavioral Neuroscience* (6% in total, 13% in 2021).

**Table 5. Relative frequency of confidence interval use in articles with p-values.**

| Journal | 2010 | 2011 | 2012 | 2013 | 2014 | 2015 | 2016 | 2017 | 2018 | 2019 | 2020 | 2021 | Total |
|---|---|---|---|---|---|---|---|---|---|---|---|---|---|
| Behavioral Neuroscience | .01 | .01 | .01 | .05 | .07 | .11 | .05 | .17 | .06 | .08 | .11 | .13 | .06 |
| Depression & Anxiety | .33 | .30 | .33 | .40 | .40 | .45 | .48 | .43 | .57 | .49 | .54 | .57 | .44 |
| Frontiers in Psychology | .15 | .08 | .10 | .10 | .12 | .16 | .21 | .25 | .25 | .29 | .29 | .32 | .25 |
| J. Abnormal Psychology | .22 | .31 | .24 | .27 | .28 | .42 | .32 | .50 | .52 | .46 | .41 | .65 | .37 |
| J. Child Psych. & Psychiatry | .27 | .25 | .36 | .40 | .44 | .42 | .42 | .48 | .50 | .45 | .53 | .55 | .42 |
| J. Family Psychology | .22 | .22 | .22 | .28 | .24 | .36 | .32 | .35 | .43 | .44 | .35 | .41 | .32 |
| J. Management | .25 | .14 | .13 | .14 | .18 | .41 | .34 | .35 | .39 | .35 | .44 | .56 | .32 |
| Pers. and Social Psych. Bull. | .19 | .24 | .25 | .42 | .55 | .87 | .90 | .86 | .80 | .88 | .89 | .90 | .63 |
| PLoS ONE | .11 | .12 | .16 | .20 | .21 | .24 | .24 | .26 | .29 | .31 | .34 | .34 | .26 |
| Psychological Medicine | .44 | .48 | .47 | .47 | .44 | .49 | .46 | .52 | .44 | .47 | .51 | .47 | .47 |
| Psychology & Aging | .12 | .20 | .15 | .16 | .24 | .20 | .28 | .24 | .36 | .45 | .29 | .44 | .25 |
| Psychophysiology | .09 | .06 | .09 | .09 | .09 | .13 | .20 | .16 | .22 | .29 | .33 | .38 | .19 |
| Total | .20 | .18 | .18 | .21 | .22 | .25 | .27 | .29 | .30 | .33 | .34 | .35 | .28 |

**Table 6. Relative frequency of power values or mentions of power analysis by journal and year in articles with p-values.**

| Journal | 2010 | 2011 | 2012 | 2013 | 2014 | 2015 | 2016 | 2017 | 2018 | 2019 | 2020 | 2021 | Total |
|---|---|---|---|---|---|---|---|---|---|---|---|---|---|
| Behavioral Neuroscience | .00 | .00 | .00 | .01 | .00 | .03 | .00 | .02 | .06 | .06 | .04 | .13 | .02 |
| Depression & Anxiety | .04 | .03 | .06 | .05 | .09 | .08 | .07 | .07 | .08 | .07 | .08 | .14 | .07 |
| Frontiers in Psychology | .04 | .02 | .02 | .03 | .04 | .04 | .05 | .07 | .09 | .12 | .12 | .12 | .09 |
| J. Abnormal Psychology | .02 | .03 | .04 | .05 | .08 | .04 | .07 | .05 | .08 | .17 | .15 | .12 | .07 |
| J. Child Psych. & Psychiatry | .07 | .04 | .04 | .09 | .06 | .07 | .10 | .13 | .14 | .14 | .12 | .11 | .09 |
| J. Family Psychology | .04 | .03 | .03 | .05 | .06 | .04 | .05 | .06 | .05 | .10 | .08 | .13 | .06 |
| J. Management | .00 | .05 | .10 | .04 | .00 | .00 | .02 | .01 | .01 | .00 | .09 | .03 | .02 |
| Pers. and Social Psych. Bull. | .02 | .01 | .00 | .02 | .05 | .22 | .37 | .34 | .53 | .63 | .63 | .62 | .27 |
| PLoS ONE | .02 | .04 | .05 | .06 | .06 | .08 | .07 | .08 | .10 | .11 | .12 | .12 | .09 |
| Psychological Medicine | .07 | .06 | .06 | .07 | .06 | .06 | .05 | .10 | .03 | .08 | .07 | .05 | .06 |
| Psychology & Aging | .05 | .01 | .03 | .05 | .06 | .05 | .06 | .08 | .23 | .23 | .26 | .33 | .11 |
| Psychophysiology | .03 | .02 | .04 | .04 | .05 | .04 | .05 | .12 | .12 | .19 | .26 | .30 | .11 |
| Total | .03 | .03 | .04 | .05 | .05 | .06 | .07 | .09 | .10 | .12 | .13 | .13 | .09 |

## Power analysis

The use of power analysis in articles that contain extractable p-values has more than quadrupled, from 3% in 2010 to 13% in 2021, but even so heavily varies between journals. In every journal, an increase of power analytical concepts is observed. Table 6 shows the relative amount of articles that report any power value by journal and year. Whereas comparatively few articles published by the *Personality and Social Psychology Bulletin* report the use of power analysis before 2015, the proportion of articles with power analysis peaked at 63% in 2019 and 2020. Also, articles in *Psychology and Aging* show a strong increase in reports of power analyses, with 33% of all articles in 2021. Lowest increase rates are observed for the *Journal of Management* and *Psychological Medicine*. The lowest overall rates of power analysis are observed for *Behavioral Neuroscience* and the *Journal of Management* (both with 2% overall rates).

Since *JATSdecoder* does not discriminate between a priori and post hoc power values, Table 7 shows the absolute and relative frequencies of the first detected power value of articles that report power as categorized values. More than half of the 3,094 articles that report power values (56%) seem to have made use of the a priori set standard value of .8. The values that are either .9 or .95 (10% each) could have been listed as a priori or a posteriori values. Since studies with power below .8 would not be very meaningful, it is probable that all values that fall within an interval were calculated post hoc.

## Bayesian statistics

The overall rate of articles mentioning Bayesian inferential methods (extractions of the Bayesian information criterion have been removed) has quadrupled from 1% in 2010 to 4% in 2021. Table 8 shows the relative amount of articles that mention Bayesian inferential methods by journal and year. The highest rate is observed in the *Journal of Management* in 2015 (26%),

**Table 7. Absolute (h(x)) and relative (f(x)) frequency distribution of the first detected and categorized power value per article.**

| Power | [0; .8) | .8 | (.8; .85) | .85 | (.85; .9) | .9 | (.9; .95) | .95 | (.95; 1] | Total |
|---|---|---|---|---|---|---|---|---|---|---|
| h(x) | 443 | 2,335 | 81 | 69 | 65 | 411 | 81 | 429 | 274 | 4,188 |
| f(x) | .11 | .56 | .02 | .02 | .02 | .10 | .02 | .10 | .07 | 1 |

**Table 8. Relative frequency of application of Bayesian inferential statistics by journal and year.**

| Journal | 2010 | 2011 | 2012 | 2013 | 2014 | 2015 | 2016 | 2017 | 2018 | 2019 | 2020 | 2021 | Total |
|---|---|---|---|---|---|---|---|---|---|---|---|---|---|
| Behavioral Neuroscience | .00 | .00 | .00 | .01 | .02 | .00 | .00 | .08 | .04 | .04 | .04 | .07 | .02 |
| Depression & Anxiety | .01 | .01 | .01 | .02 | .01 | .02 | .01 | .02 | .03 | .00 | .04 | .03 | .02 |
| Frontiers in Psychology | .02 | .01 | .05 | .04 | .04 | .05 | .05 | .04 | .05 | .05 | .05 | .04 | .04 |
| J. Abnormal Psychology | .02 | .02 | .00 | .01 | .01 | .03 | .07 | .04 | .11 | .10 | .15 | .12 | .05 |
| J. Child Psych. & Psychiatry | .00 | .00 | .00 | .02 | .03 | .03 | .01 | .02 | .00 | .03 | .04 | .03 | .02 |
| J. Family Psychology | .00 | .02 | .00 | .02 | .01 | .02 | .03 | .01 | .01 | .02 | .04 | .01 | .02 |
| J. Management | .00 | .00 | .00 | .00 | .00 | .26 | .02 | .03 | .02 | .06 | .00 | .03 | .04 |
| Pers. and Social Psych. Bull. | .00 | .00 | .00 | .00 | .01 | .03 | .03 | .01 | .04 | .04 | .07 | .09 | .02 |
| PLoS ONE | .03 | .02 | .02 | .03 | .02 | .02 | .04 | .03 | .05 | .05 | .05 | .04 | .04 |
| Psychological Medicine | .01 | .01 | .01 | .01 | .00 | .00 | .01 | .02 | .03 | .03 | .04 | .06 | .02 |
| Psychology & Aging | .03 | .01 | .01 | .02 | .01 | .04 | .07 | .10 | .12 | .17 | .13 | .17 | .07 |
| Psychophysiology | .00 | .01 | .01 | .02 | .00 | .03 | .03 | .08 | .04 | .11 | .14 | .13 | .05 |
| Total | .01 | .01 | .02 | .02 | .02 | .03 | .04 | .04 | .04 | .05 | .05 | .04 | .04 |

which published a special issue with the title *Bayesian Probability and Statistics in Management Research* in February. Still, great differences can be observed between the journals. While in 2021 17% of all empirical research articles published by *Psychology and Aging* use inferential Bayesian methods, the rate in the *Journal of Family Psychology* is 1%.

## Preregistration and multiverse analyses

The 'preregistration revolution' called for by Nosek et al. [37] has not yet taken place. A clear distinction of preregistered and registered reports and protocols cannot be made. In six of the 12 journals, a text search in the title and abstract with the regular expression pattern 'registered report|registered replication|registered stud[yi]' identified only 82 articles (0.14%), that were preregistered or registered. Table 9 shows the absolute frequency of detected preregistered or registered reports by journal and year. The first preregistered reports were published by *Frontiers in Psychology* in 2013. Overall, *PLoS One* has published most preregistered psychological reports or protocols, with 40 articles. A text search with the pattern 'replicat' in the title and abstract reveals that 25 of the 82 articles (30%) are replications. In total, the same search task

**Table 9. Absolute frequency of preregistered or registered reports by journal and year.**

| Journal | 2010 | 2011 | 2012 | 2013 | 2014 | 2015 | 2016 | 2017 | 2018 | 2019 | 2020 | 2021 | Total |
|---|---|---|---|---|---|---|---|---|---|---|---|---|---|
| Behavioral Neuroscience | 0 | 0 | 0 | 0 | 0 | 0 | 0 | 0 | 0 | 0 | 0 | 0 | 0 |
| Depression & Anxiety | 0 | 0 | 0 | 0 | 0 | 0 | 0 | 0 | 0 | 0 | 0 | 0 | 0 |
| Frontiers in Psychology | 0 | 0 | 0 | 2 | 1 | 2 | 1 | 0 | 1 | 3 | 11 | 7 | 28 |
| J. Abnormal Psychology | 0 | 0 | 0 | 0 | 0 | 0 | 0 | 0 | 1 | 0 | 0 | 1 | 2 |
| J. Child Psych. & Psychiatry | 0 | 0 | 0 | 0 | 0 | 0 | 0 | 0 | 0 | 0 | 0 | 0 | 0 |
| J. Family Psychology | 0 | 0 | 0 | 0 | 0 | 0 | 0 | 0 | 0 | 0 | 0 | 0 | 0 |
| J. Management | 0 | 0 | 0 | 0 | 0 | 0 | 0 | 0 | 0 | 0 | 0 | 0 | 0 |
| Pers. and Social Psych. Bull. | 0 | 0 | 0 | 0 | 0 | 0 | 0 | 0 | 1 | 2 | 0 | 2 | 5 |
| PLoS ONE | 0 | 0 | 0 | 0 | 0 | 0 | 0 | 1 | 3 | 6 | 8 | 22 | 40 |
| Psychological Medicine | 0 | 0 | 0 | 0 | 0 | 0 | 0 | 0 | 0 | 0 | 0 | 0 | 0 |
| Psychology & Aging | 0 | 0 | 0 | 0 | 0 | 0 | 0 | 0 | 0 | 1 | 0 | 1 | 2 |
| Psychophysiology | 0 | 0 | 0 | 0 | 0 | 0 | 0 | 0 | 0 | 0 | 2 | 3 | 5 |
| Total | 0 | 0 | 0 | 2 | 1 | 2 | 1 | 1 | 6 | 12 | 21 | 36 | 82 |

identified 1,992 (3.4%) replications in the entire collection of articles. Thus, among the replications, preregistration is more common, although still quite rare (1.3%).

Multiverse analyses are performed even less frequently than preregistrations in the literature analyzed here. A search task on the title, abstract, and extracted statistical methods with the regular expression '[Mm]ulti[- ]*verse' yielded only ten articles mentioning multiverse analyses. The first four mentions are observed in 2018, one of which is is a non-excluded theoretical paper by Gelman [5] 'The Failure of Null Hypothesis Significance Testing When Studying Incremental Changes, and What to Do About It'. In each of the following three years, two articles were identified that used multiverse analyses.

## Correction for multiple testing

Although most of the articles that contain p-values use multiple testing, only 24% report the use of a corrected $\alpha$ level. There is great variability between journals regarding to the use of procedures that correct the $\alpha$ level for multiple testing. Whereas half of all included articles with p-values by *Behavioral Neuroscience* report the use of corrections for multiple testing, with a peak of 62% in 2019, only 2% of all articles by the *Journal of Management* report a correction of $\alpha$ levels. Except for *PLoS ONE* and *Frontiers in Psychology*, all other journals contain an increased amount of articles with multiple test corrections since 2016, although most changes are rather weak (see Table 10).

Fig 4 shows the relation of the relative amount of articles that report the use of correction procedures for multiple testing and the number of extracted p-values from text for every journal. In general, the proportion of articles that make use of $\alpha$ level correction procedures increases with the number of reported p-values in almost every journal. Except for articles with more than 80 p-values, articles published in *Behavioral Neuroscience* make the most intensive use of $\alpha$ level correction procedures, probably because they are an implemented standard in fMRI data analysis software. The lowest correction rates are observed for the *Journal of Management*, which, at the same time, has the lowest maximum number of 74 extractable p-values within text. Seven out of eight articles with more than 200 p-values, which are published by *PLoS One* and *Frontiers in Psychology*, report the use of $\alpha$ level corrections.

## Test direction

Although the extraction heuristics showed to have a high accuracy, an explicit report of one-tailed testing is found in only 4% of all articles. At least one one-tailed test is detected in 5% of all articles published in and before 2015 and in 3% of all articles published after 2015 (see Table 1). Table 11 shows the relative frequency of detected test direction by journal. The highest amount of articles using one-tailed tests is found in *Psychophysiology* (8%).

## Estimated sample size

The estimated sample sizes vary greatly between journals. Table 12 displays the journal specific median and .75-quanile of the estimated sample sizes in and before 2015 and after 2015 as well as the corresponding increase factors. Except for *Behavioral Neuroscience*, the median sample sizes have increased in every journal. The .75-quantiles of sample sizes have also increased in every journal. In articles published after 2015 in *Depression & Anxiety* and *Psychological Medicine*, more than 25% of all estimated sample sizes are greater than 2,000.

A more detailed overview of the median estimated sample sizes by journal and year, as well as global medians, is presented in Table 13. More than half of all annual estimated sample sizes in *Behavioral Neuroscience* are below 75 and in *Psychophysiology* below 85. All other journals

**Table 10. Relative frequency of at least one detected multiple test correction procedure by journal and year in articles with p-values.**

| Journal | 2010 | 2011 | 2012 | 2013 | 2014 | 2015 | 2016 | 2017 | 2018 | 2019 | 2020 | 2021 | Total |
|---|---|---|---|---|---|---|---|---|---|---|---|---|---|
| Behavioral Neuroscience | .48 | .48 | .43 | .52 | .48 | .55 | .50 | .46 | .61 | .62 | .59 | .36 | .50 |
| Depression & Anxiety | .09 | .21 | .16 | .16 | .24 | .19 | .16 | .21 | .28 | .27 | .20 | .26 | .20 |
| Frontiers in Psychology | .21 | .27 | .26 | .28 | .27 | .26 | .25 | .26 | .25 | .23 | .21 | .18 | .23 |
| J. Abnormal Psychology | .19 | .14 | .16 | .15 | .14 | .18 | .15 | .16 | .21 | .28 | .28 | .31 | .19 |
| J. Child Psych. & Psychiatry | .16 | .18 | .17 | .24 | .17 | .13 | .25 | .24 | .16 | .19 | .23 | .25 | .20 |
| J. Family Psychology | .16 | .07 | .03 | .08 | .06 | .10 | .12 | .08 | .09 | .12 | .07 | .10 | .09 |
| J. Management | .00 | .00 | .00 | .06 | .02 | .02 | .04 | .03 | .00 | .01 | .00 | .03 | .02 |
| Pers. and Social Psych. Bull. | .07 | .09 | .05 | .04 | .06 | .05 | .08 | .08 | .09 | .06 | .10 | .11 | .07 |
| PLoS ONE | .33 | .34 | .34 | .35 | .33 | .31 | .29 | .27 | .23 | .23 | .20 | .19 | .26 |
| Psychological Medicine | .19 | .23 | .19 | .23 | .22 | .29 | .33 | .22 | .30 | .28 | .35 | .31 | .26 |
| Psychology & Aging | .13 | .11 | .15 | .16 | .16 | .14 | .12 | .16 | .20 | .18 | .22 | .20 | .16 |
| Psychophysiology | .28 | .28 | .31 | .36 | .40 | .31 | .34 | .38 | .38 | .46 | .45 | .46 | .37 |
| Total | .22 | .26 | .27 | .30 | .29 | .26 | .26 | .25 | .24 | .23 | .21 | .19 | .24 |

show to have much higher median sample sizes with a maximum of 732 in the Journal of Child Psychology & Psychiatry in 2021.

Table 14 shows the median of the estimated sample sizes for articles with and without a repeated measures analysis (RMA) and the proportions of articles with RMA. The partitioning

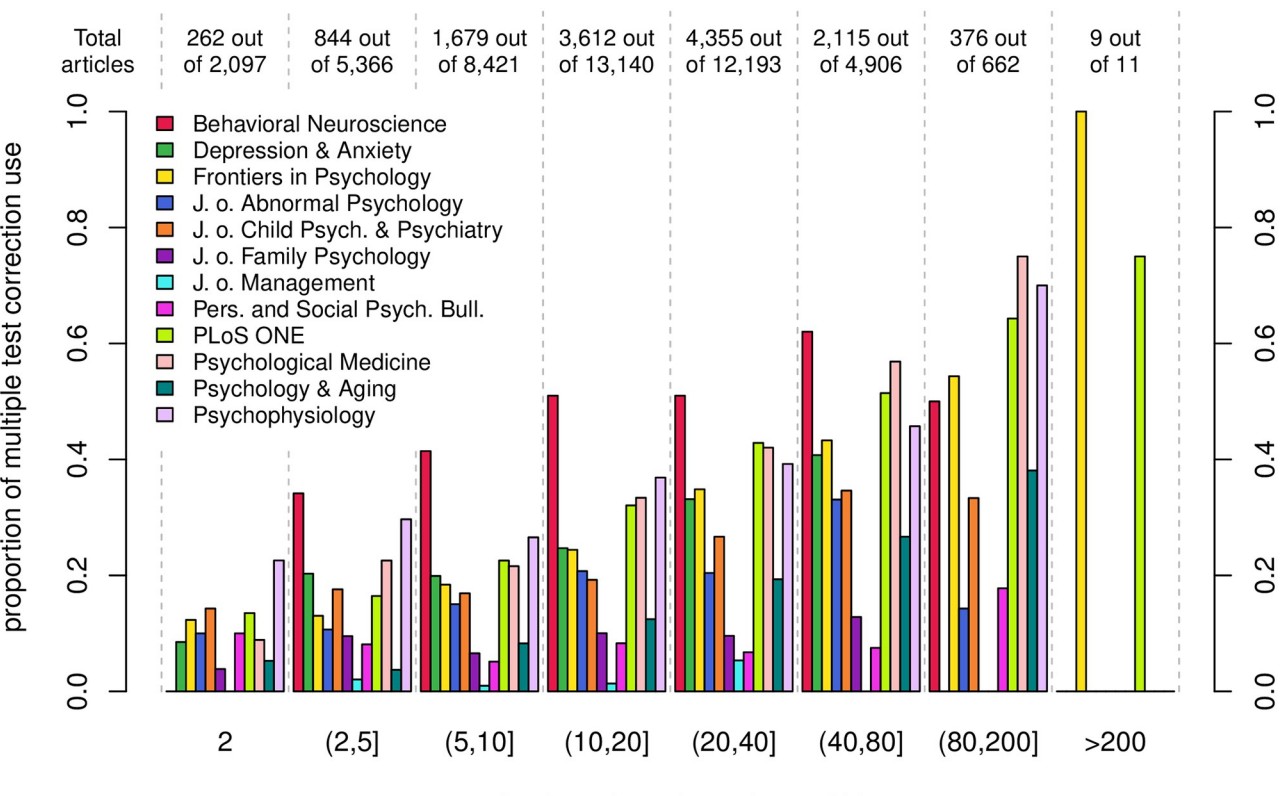

**Fig 4. Absolute frequencies of articles with correction procedures by categorized number of p-values (numbers on top) and journal-wise relation of number of extracted p-values from text and use of correction procedures for multiple testing (bars).**

**Table 11. Relative frequencies of detected test direction by journal.**

| Journal | no detection | one and two sided | one sided | two sided |
|---|---|---|---|---|
| Behavioral Neuroscience | .85 | .02 | .03 | .10 |
| Depression & Anxiety | .79 | .01 | .01 | .19 |
| Frontiers in Psychology | .89 | .01 | .03 | .07 |
| J. Abnormal Psychology | .88 | .01 | .02 | .08 |
| J. Child Psych. & Psychiatry | .86 | .01 | .02 | .11 |
| J. Family Psychology | .94 | .00 | .02 | .04 |
| J. Management | .80 | .03 | .04 | .13 |
| Pers. and Social Psych. Bull. | .89 | .01 | .03 | .06 |
| PLoS ONE | .84 | .01 | .02 | .13 |
| Psychological Medicine | .81 | .01 | .01 | .17 |
| Psychology & Aging | .91 | .01 | .03 | .05 |
| Psychophysiology | .80 | .03 | .05 | .12 |
| Total | .86 | .01 | .03 | .11 |

was performed with a search in the extracted statistical methods using the regular expression *'repeated measure|cross lagged|panel |longitud'*. While the proportion of articles with RMA decreased from 25% to 16%, the median of the estimated sample size increased for both types of designs (68 to 89 in articles with RMA, 129 to 232 in articles without RMA). Both, the absolute and relative increase in median estimated sample size is higher for articles without measurement repetition.

## Country of origin

Fig 5 shows the change in relative frequency of country involvement for articles published before and after 2015 for the most frequently detected countries of origin after 2015. Whereas the relative amount of articles from the United States has decreased most (26.6% to 19.6%), the relative amount of articles from China has nearly doubled (4.2% to 7.9%.) In relative terms, Norway, Poland and Portugal, three comparably small distributors in and before 2015, have increased their output the most.

**Table 12. Median and .75-quantile of estimated sample size before and in 2015 and after 2015 by journal.**

| Journal | median | | | .75-quantile | | |
|---|---|---|---|---|---|---|
| | ≤ 2015 | > 2015 | factor | ≤ 2015 | > 2015 | factor |
| Behavioral Neuroscience | 58 | 56 | 0.97 | 116.75 | 134 | 1.15 |
| Depression & Anxiety | 217 | 390.5 | 1.8 | 1164 | 2068.25 | 1.78 |
| Frontiers in Psychology | 77 | 189.5 | 2.46 | 179 | 525 | 2.93 |
| J. Abnormal Psychology | 207.5 | 262 | 1.26 | 726.75 | 900 | 1.24 |
| J. Child Psych. & Psychiatry | 265.5 | 461.5 | 1.74 | 1186.5 | 1920.75 | 1.62 |
| J. Family Psychology | 246 | 300 | 1.22 | 593 | 628 | 1.06 |
| J. Management | 243 | 304 | 1.25 | 562.5 | 800.75 | 1.42 |
| Pers. and Social Psych. Bull. | 161 | 356 | 2.21 | 305 | 900 | 2.95 |
| PLoS ONE | 82 | 170 | 2.07 | 317.25 | 724.5 | 2.28 |
| Psychological Medicine | 311 | 326 | 1.05 | 2075 | 2705.25 | 1.3 |
| Psychology & Aging | 156 | 201 | 1.29 | 448 | 994 | 2.22 |
| Psychophysiology | 61 | 74 | 1.21 | 129 | 143 | 1.11 |

**Table 13. Median estimated sample size by journal and year.**

| Journal | 2010 | 2011 | 2012 | 2013 | 2014 | 2015 | 2016 | 2017 | 2018 | 2019 | 2020 | 2021 | Total |
|---------|------|------|------|------|------|------|------|------|------|------|------|------|-------|
| Behavioral Neuroscience | 56 | 72 | 47 | 60 | 55 | 64 | 74 | 45 | 59 | 50 | 56 | 64 | 57 |
| Depression & Anxiety | 260 | 151 | 274 | 273 | 205 | 240 | 368 | 182 | 344 | 334 | 474 | 519 | 285 |
| Frontiers in Psychology | 46 | 53 | 52 | 73 | 78 | 95 | 114 | 150 | 136 | 175 | 228 | 272 | 149 |
| J. Abnormal Psychology | 208 | 214 | 202 | 237 | 127 | 228 | 199 | 321 | 162 | 198 | 571 | 462 | 229 |
| J. Child Psych. & Psychiatry | 256 | 190 | 285 | 244 | 263 | 404 | 399 | 286 | 208 | 648 | 518 | 732 | 365 |
| J. Family Psychology | 297 | 230 | 210 | 228 | 279 | 288 | 285 | 269 | 305 | 247 | 331 | 306 | 268 |
| J. Management | 184 | 389 | 194 | 282 | 221 | 374 | 379 | 300 | 241 | 329 | 373 | 301 | 287 |
| Pers. and Social Psych. Bull. | 132 | 141 | 132 | 183 | 170 | 245 | 234 | 317 | 356 | 328 | 406 | 557 | 233 |
| PLoS ONE | 49 | 59 | 76 | 77 | 98 | 106 | 114 | 127 | 166 | 173 | 201 | 282 | 127 |
| Psychological Medicine | 192 | 248 | 490 | 318 | 421 | 258 | 224 | 273 | 530 | 410 | 419 | 331 | 318 |
| Psychology & Aging | 152 | 181 | 111 | 150 | 254 | 186 | 189 | 175 | 172 | 481 | 164 | 295 | 175 |
| Psychophysiology | 51 | 52 | 61 | 52 | 79 | 73 | 72 | 59 | 71 | 81 | 84 | 80 | 68 |
| Total | 114 | 97 | 97 | 100 | 108 | 119 | 129 | 148 | 165 | 190 | 220 | 271 | 151 |

Fig 6 shows the relative frequency of article origin from WEIRD and non-WEIRD countries of origin for the publication period focused on here. The amount of articles from non-WEIRD countries has steadily increased within the last 12 years. Whereas in 2010 a relatively small amount of articles (12%) was published or co-published by authors from non-WEIRD countries, non-WEIRD country involvement increased to 43.7% in 2020. In 2021, 21.4% of the articles with extractable country were published solely by authors from non-WEIRD countries, whereas 56.3% were published solely by authors from WEIRD countries.

## Discussion

This study contributes to the ongoing discussion about the replication crisis in psychology and is the first application of *JATSdecoder* to gain detailed insights into the research practice of a subject. Surprisingly, most of the long-proposed methodological reforms listed above are implemented only hesitantly or not at all.

In psychological research, the use of nil-null hypothesis testing is pervasive. Interval hypotheses are almost never tested. The global median of the reported number of p-values per study is 13. The analysis supports Meehl's and Cohen's statement about the predictably high rejection rate of point-null hypotheses. Regardless of the type of test and sample size, 68% of the recalculated p-values are significant at $\alpha = .05$. In half of the articles, more than two-thirds of the extracted and three-quarters of the recalculated p-values are below.05.

Given the high rejection rates of null hypotheses and the evidence from the literature that point null hypotheses are all too easily rejected, a critical examination of the common practice of undirected nil-null hypothesis testing seems necessary. In practice, thresholds other than

**Table 14. Median estimated sample sizes in articles with and without repeated measures analysis and proportion of articles with repeated measures design.**

| Design type | ≤2015 | >2015 | $\delta$ | 99.9% CI for $\delta$ |
|-------------|-------|-------|----------|----------------------|
| with repeated measures | 68 | 89 | 21 | [13; 27] |
| without repeated measures | 129 | 232 | 103 | [88; 116] |
| proportion with repeated measures | 0.25 | 0.16 | -0.085 | [-0.103; -0.068] |

Note: CIs for $\delta$ represent bootstrap confidence intervals for differences in medians based on 20,000 resamples and a confidence interval for difference in proportions.

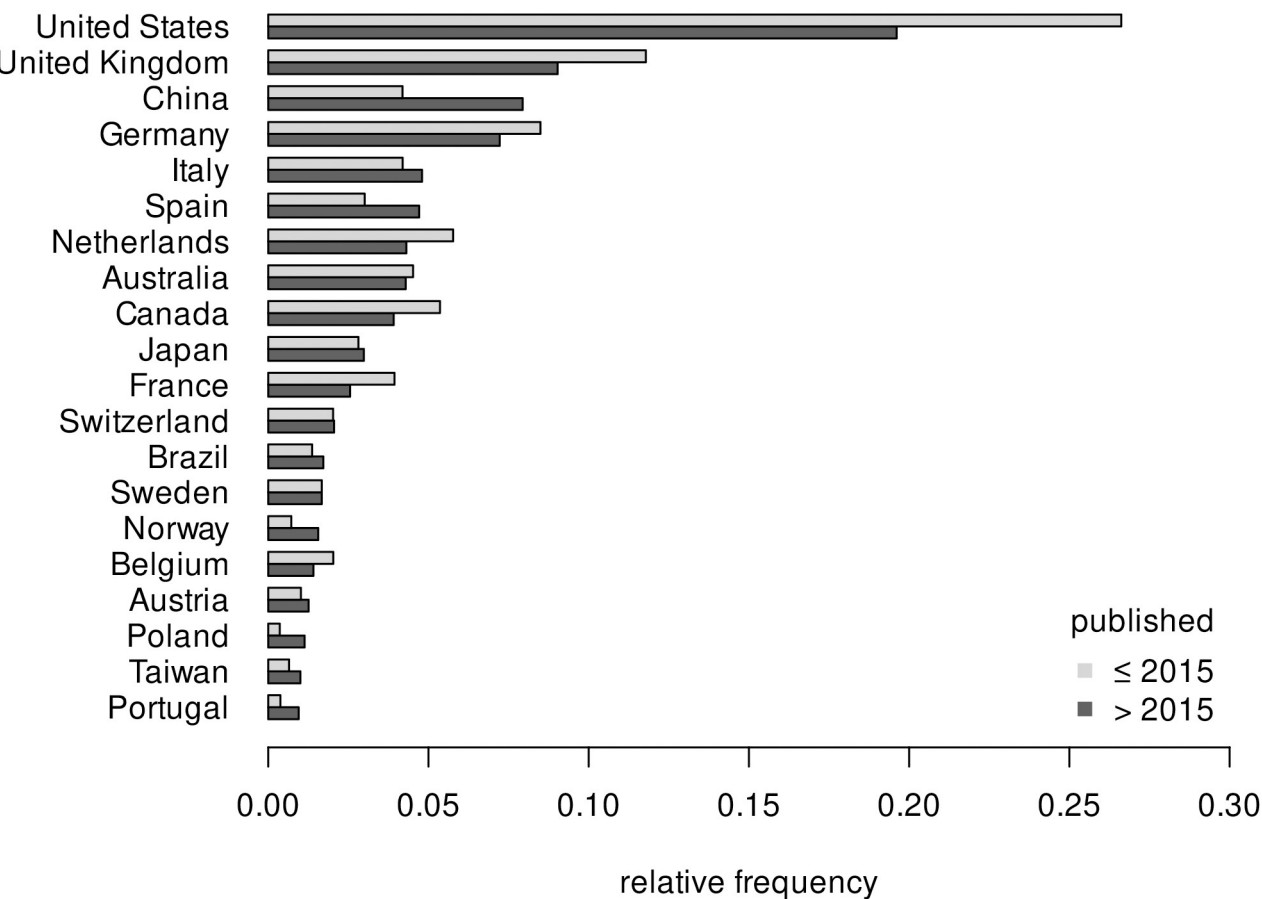

**Fig 5. Change in relative country involvement before and after 2015.**

zero would need to be discussed in the community and adjusted over time, but could serve as an indicator of relevance or quality in a particular research area or domain.

Departing from the ritualized nil-null hypothesis test would certainly be associated with a higher rate of 'unsuccessful' studies, but at the same time the focus would shift to the relevance of the observed effects and the resilience of the underlying theories. A threshold-based approach can also produce non-significant results as strong evidence for a theory when non-zero effects are tested with sample sizes that yield high power. In addition, a significant result would then have to be considered as a refutation or limitation of a theory, which is not the case with non-significant results in low-powered studies.

The methodological study features focused on here vary greatly between journals. Some of these features have changed within the last 12 years. Certainly the underlying effects for the observed changes are multicausal and dynamic. Besides general concerns about reproducibility and a general greater awareness of methodological issues, the causes may be strongly driven by journal requirements, the adoption of changing standards by researchers, and editors and reviewers enforcing these standards.

The reporting of correction procedures for multiple testing has decreased in articles published after 2015. Although it is not indicated to strictly correct for the cumulation of $\alpha$ errors of every test carried out in a study [53], the large number of hypothesis tests per study indicates that at least some sort of $\alpha$ error adjustment is required in many cases. The simple call to set

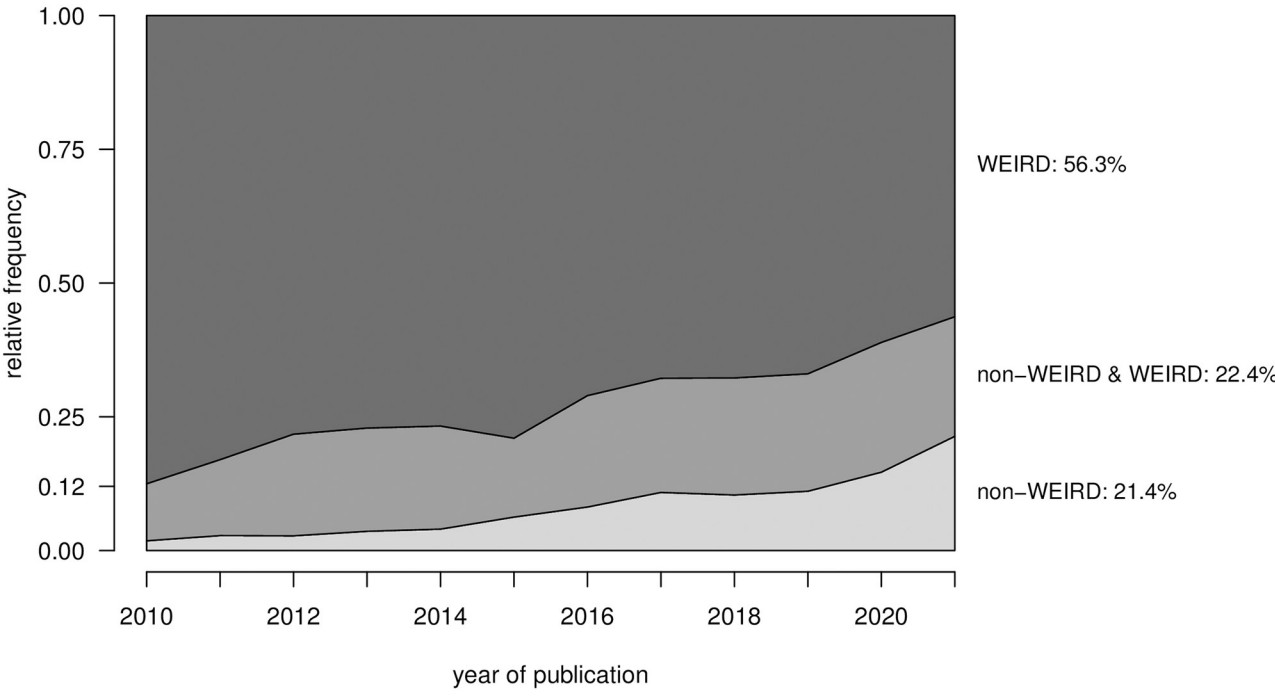

**Fig 6. Change in WEIRD and non-WEIRD country of origin over time.**

the $\alpha$ level lower than.05 is not followed. Instead, $\alpha$ levels greater than.05 were found more frequently than values below.05. Although corrections for multiple testing and lower $\alpha$ errors are associated with lower power, their use should be considered more frequently in psychology.

Standard effect sizes are increasingly often reported in nearly every journal, which is beneficial for meta-analyses that accumulate effect sizes. Also, the reporting of confidence intervals has steadily increased, from 20% in all articles in 2010 to 35% in 2021. In the journal *Personality and Social Psychology Bulletin*, reporting effect sizes and confidence intervals has become the general standard. This is not surprising since the publication guidelines require their reporting regardless of the significance level. However, it remains unclear whether confidence intervals are mainly interpreted in terms of their width or as a simple substitute for a significance test, which would not be any information gain.

The estimated sample sizes have increased within the examined period from a median of 105 in articles published in and before 2015 to 190 in articles published after 2015. The trend toward larger samples is observed in both repeated-measures and single-measures designs. Assuming that the effect sizes of interest have not changed, this is an indication of increased test power and a good sign. However, it is not yet clear whether this effect is related to the more frequent use of power analyses or to the increasing ease of obtaining samples through online surveys or other sources.

Although several journals request a power analysis in their current guidelines, realized sample sizes are rarely justified with power analysis. The proportion of reported power analyzes has increased from only 3% in 2010 and 2011 to 13% in 2020 and 2021. This could be due to the high cost that a power analysis would indicate for collecting a sample needed to demonstrate a realistically estimated effect size. An intriguing question would be with what effect sizes the reported power analyses were conducted, and whether there is a stringent relationship with the realized sample sizes.

Besides the fact that any good theory implies the direction of an effect, directed testing represents a simple and cost-saving way to increase the power of a statistical test. Because one-tailed testing is generally a rare practice, doubts about post-hoc decisions on test sidedness (which surely is a questionable research practice) can be dispelled by preregistration.

None of the the journals analyzed here requires preregistration, and preregistration is rarely used in practice. This would be particularly desirable for confirmatory studies, as preregistration increases the credibility of results by minimizing questionable research practices. However, to ensure that non-significant results are included in the literature, preregistered reports should be peer reviewed in advance to data collection and published regardless of the outcome.

The origin of psychological research has turned to a less WEIRD one, mostly due to the many recent publications from China. It is reasonable to conclude, that also the origin of samples has diversified. However, it remains to be investigated whether the samples have become more multicultural on the article level and whether other sample characteristics have changed, for example whether students are less over-represented in the samples. Digital questionaires and worker markets have made it very easy to invite international participants to surveys. How much psychological research is based international samples nowadays remains an open question for future research. From a population-based perspective, research from African and Latin American countries is still severely underrepresented, which may be due to less research funding in those countries.

Here, mainly global comparisons were performed. More detailed analyses of specific subgroups of studies and correlations of study characteristics are still pending. Study characteristics in all open access journals that are part of the PubMed Central database can be easily downloaded or analyzed via an interactive web application, accessible at: www.scianalyzer.com. It allows identification and analysis of individual article selections by journal, topic, author, affiliation, time ranges and study features extracted with *JATSdecoder*. Further, the data export provided enables individual analyses of the study properties for selected articles.

## Limitations

Although a rather big sample of articles was included for this analysis, the article collection represents a selective sample by highly renowned journals with high standards. The results should therefore not be generalized to other journals or to psychology in general. As no weighting has been performed, the observed global changes are heavily influenced by the high number of articles by both here included open access journals, that supply 77% of the analyzed articles. The selection process for original research articles may have resulted in false positive and negative inclusions. However, this should have little to no impact on the reported results.

The *JATSdecoder* algorithms are precise, but not error-free. Extraordinary or only implicitly inferred study features cannot be extracted, resulting in a negative bias. Additionally, journal- and time period-specific extraction biases may have occurred, limiting the comparability of extracted study chararacteristics. Since statistical results within tables were not extracted, the total number of test results is certainly higher. Since the conversion of special characters in PDF documents can be error-prone and results reported in tables and figures are not extracted with *JATSdecoder* the frequencies of reported test results and standard effect sizes may have been underestimated.

## Acknowledgments

The author thanks Marcella Dudley for the linguistic revision of the manuscript.

## Author Contributions

**Conceptualization:** Ingmar Böschen.

**Data curation:** Ingmar Böschen.

**Formal analysis:** Ingmar Böschen.

**Methodology:** Ingmar Böschen.

**Software:** Ingmar Böschen.

**Writing – original draft:** Ingmar Böschen.

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
