## [Decision Letter · Decision Letter 0]

12 Jul 2022

PONE-D-22-14556Changes in methodological study characteristics in psychology between 2010-2021PLOS ONE

Dear Dr. Böschen,

Thank you for submitting your manuscript to PLOS ONE. After careful consideration, we feel that it has merit but does not fully meet PLOS ONE’s publication criteria as it currently stands. Therefore, we invite you to submit a revised version of the manuscript that addresses the points raised during the review process.

The reviewers see many positive aspects in your work, but also raise several issues. In particular, all three reviewers recommend a series of significant revisions on the statistical analyses performed. Please carefully address all of the reviewers' points before resubmitting your manuscript.

We look forward to receiving your revised manuscript.

Kind regards,

Enrico Toffalini, Ph.D

Academic Editor

PLOS ONE

Journal Requirements:

"This research was financed by a doctoral grant awarded by the Department of Psychological Methods and Statistics, Institute of Psychology, University Hamburg, Germany."

Reviewers' comments:

Reviewer's Responses to Questions

**Comments to the Author**

1. Is the manuscript technically sound, and do the data support the conclusions?

Reviewer #1: Partly

Reviewer #2: Yes

Reviewer #3: Partly

2. Has the statistical analysis been performed appropriately and rigorously? 

Reviewer #1: No

Reviewer #2: Yes

Reviewer #3: No

3. Have the authors made all data underlying the findings in their manuscript fully available?

Reviewer #1: Yes

Reviewer #2: Yes

Reviewer #3: Yes

4. Is the manuscript presented in an intelligible fashion and written in standard English?

Reviewer #1: Yes

Reviewer #2: Yes

Reviewer #3: Yes

5. Review Comments to the Author

Reviewer #1: “Changes in methodological study characteristics in psychology between 2010-2021” reports a series of statistics regarding several characteristics of statistical analyses and techniques presented in a large body of published empirical research in psychology. Results are gathered employing a text mining tool that allows extracting granular information from the original published papers.

Is the manuscript technically sound, and do the data support the conclusions? PARTLY

The author lists a series of statistical practices that have been pointed out in the literature as putative causes of the difficulty to replicate results in psychology. Those issues are briefly and somehow uncritically reviewed and linked to empirical indicators extractable from the body of evidence analyzed in the manuscript. I find both the list of issues and their link with the data too weak.

a) Low power = small samples = high uncertainty

Statistical power is undoubtedly as positive property of a statistically analyzed study. As the author mentioned, however, so is noisy measurements. Also, power is linked to sample size and expected effect sizes, and it should be estimated before running a study. Of all these important points, the authors gather data only on frequency of a priori power analysis and sample size.

It would be interesting to see whether study with a priori power analysis showed greater power than study without an a priori power analysis (given the observed effect sizes) or at least whether study with an a priori power analysis employed a larger sample as compared with studies that do not run a power analysis. This would help to really understand the consequences of adherence to “best practices”. We all assume that best practices are beneficial for the quality of research, but nobody brings evidence that they are. A study like this one is an opportunity to check the effects of these practices.

Coherently, I was expecting the author to gather information about reporting standardized effect size indices. It would be interesting to show effect size increased over time, and if there is a negative correlation with the sample size, as the use of power analysis would imply.

No mention of the level of measurements reliability in the analyzed paper can be found in the results section, even thought the author mentions this property as an important quality of research.

b) The undirected nil-null test scenario

The author claims that directed tests (one-sided tests) improve the quality of researching. Under some circumstances, such as in registered report, many would agree with this claim. However ,what happens in practice is that researchers willing to use one-sided tests are often accused by reviewers to capitalize on their chances to get something significant, because one sided tests are more powerful, and thus more likely to yield a significant result than two-sided tests. So, the lack of one-sided tests (5% before 2015 and 3% after) may simply reflect the more stringent adherence of reviewers to the so-called “best practices”, which usually entails asking for more stringent tests.

The other claim in this section regards the pledge to report confidence intervals. Confidence intervals, when computed based on large sample theory, contain exactly the same information than the frequentist inferential tests they come from. They just portrait the information in a different format.

Furthermore, they are even more misinterpreted that the corresponding inferential tests. Thus, reporting confidence intervals does not add or subtract anything to the quality of the research, and possibly introduce misinterpretation. Interpreting correctly the intervals, however, can be a positive addition to an article. If I recall well, Cumming’s “new statistics” was not about blindly reporting the confidence intervals, but it was about interpreting them and draw conclusions from their width. If one only uses them to check if zero is inside, a t-test/p-value would do the job just fine.

Do the author have evidence that articles that report confidence intervals also interpret them correctly, or they use them in any way which improves the evaluation of the results? Just showing that confidence intervals are reported more often now than in the past, may simply reflect changes in journals policy, without any practical effect on research quality.

c) Questionable research practices

This issue is mentioned in the introduction, but no data are reported to say something about it. This issue may be removed from the introduction, or some creative data may be found to check on its evolution over time. Number of registered reports may be analyzed, or comparisons between registered and not registered studies can be carried out on some key properties. Replications of the same experiment within the same papers will be also interesting, but I guess that this information is more difficult to extract from the data.

d) Multiple testing

The statement saying that “[..] the α level or p-values should always be adjusted/corrected for multiple tests” is too general and likely untrue. Assume I have a 3 balanced groups (A,B,C) experiment, and I registered that my theory will be supported if, and only if, A !=C and (A+C)/2!=B. If both comparisons are false, the probability of supporting my hypothesis is alpha^2, which is obviously less than alpha. If only one comparison is “true” (say the second), then my probability of supporting my hypothesis is alpha*(1-beta2), which is always less than alpha, since 0<beta<1. are="" both="" if="" true="">

Adjustment for multiple comparisons is required when one is willing to accept any statistically significant results as valid and supporting a hypothesis. So, the author should check if adjustment is more likely when no hypothesis is laid out in advance, because that is when adjustment is requirement. Showing that adjustment has increased in frequency may just be a sign that researchers are reducing their a priori hypotheses and increased their HARKing.

e) Selective samples

This section is interesting and informative.

2) Has the statistical analysis been performed appropriately and rigorously? NO

If I had the option, I would have marked this “partly”. The methods are good and fine, but I’d like to mention the necessity to add control variables to the analyses. The most striking example is the analysis of the number of p-values reported by each paper. Number of p-values may be highly correlated with number of studies reported in a paper and methods used (experimental vs correlational studies). Thus, any increase or decrease over time of the number of reported p-values per article may be uncorrelated with the inferential strategies of the researchers, but simply due to a change in format of the journals. In non-randomized designs, furthermore, number of p-values can also be correlated with the use of control variables, as a way to identify possible spurious effects. It would be ironic if more controlled studies were judged as less accurate because they present more p-values than less controlled studies.

As regards sample size (which is gathered for studying reporting it), it will make more sense to control for effect size. A small study on a large effect size may be as powerful as a large study on a small effect size. In general, sample size is correlated with the generality of the effects one is studying. Surveying detailed opinions about a product may require a much larger sample to obtain stable results than a study counting how many hemispheres there are in the human brain. As mentioned above, it would be also interesting to correlate the sample size with the presence of a priori power analysis.

In conclusions, I believe that the data collected by the author have great potential to support a solid article. However, I would suggest to elaborate more the theoretical justification and the implications of the different indicators analyzed in the paper, and find more interesting relationships among indicators that could give a more precise and insightful overview of the evolution of the field.</beta<1.>

Reviewer #2: Title: Changes in methodological study characteristics in psychology between 2010-2021

Journal: PLOS ONE

MS#: PONE-D-22-14556

I think there are several things that can be improved.

1. In the introduction, the author fails to cite the literature militating against significance testing. The author should cite that literature or explain why it is not relevant.

2. The issue of power analysis seems problematic too in that if one eschews significance testing, then why perform a power analysis to determine the sample size needed to have a good chance of getting a significant result?

3. Why is there no discussion of the fact that, as most people use confidence intervals the way they use significance tests, there is no difference in practical usage? And why is there no discussion of the fact that a confidence interval is severely limited by the fact that the researcher has no idea of the probability that the population parameter is in the interval?

4. The author does not explain why the analyses performed benefit the field.

On the positive side, the article is clearly written and in that respect conforms to PLOS ONE requirements.

Reviewer #3: This paper reports a really fascinating and I'm sure effortfully composed data set about the changes over a decade in several aspects of statistical reporting at a selection of 12 psychology journals in various fields. There are some metrics that are useful such as the general failure to adopt alphas other than .05, the prevalence of .8 as the target beta for power analysis, increases in confidence intervals, etc. There are others that are less important, such as the raw numbers of p-values per article without correcting for factors such as the number and complexity of studies per article in the journal.

In general I think the paper got stuck on some analyses that are not informative, missed some analyses that would have been informative, and needs to present and interpret its metascientific topic in a much more systematic way.

1. The paper has a not very deep appreciation of some of the criticisms of psychological methods and their limits. At times the writing is overly long and at a very basic level for the issues that are studied in the meta-science research that follows. We can assume that most researchers know the basics of hypothesis testing without needing to see the formula for a t-test, for example, and citations of sources about the controversy can be cited in passing. The Introduction needs to focus on the reason why these standards might be changing and the voices calling for change, rather than on basic-level explanations or unsupported claims. In other words, it needs to spend more time being descriptive about why change might be observed, and less being prescriptive and hinting at the author's opinions about the correct course of action. For example:

a. The Ioannidis claim that most published research findings are false is only valid under conditions of testing very low probability hypotheses, ignoring the many non-flashy studies that test well-grounded ideas. If the rate of hypothesis truth was the 10% or so required for this claim to be true, then why would selections of Registered Reports return confirmation rates of 45% or so? This suggests that hypotheses being tested in psychology are about 50% true with some attrition for type II error under high power.

Scheel, A. M., Schijen, M. R., & Lakens, D. (2021). An excess of positive results: Comparing the standard Psychology literature with Registered Reports. Advances in Methods and Practices in Psychological Science, 4(2), 25152459211007467.

see also

Wegener, D. T., Fabrigar, L. R., Pek, J., & Hoisington-Shaw, K. (2022). Evaluating research in personality and social psychology: Considerations of statistical power and concerns about false findings. Personality and Social Psychology Bulletin, 48(7), 1105-1117.

b. The discussion of power assumes that studies can be spoken of objectively as high or low powered when actually it depends critically on the effect size of interest. The assumption of a medium-sized effect as desirable and a small effect as optimum is nowhere justified.

c. The discussion of issues with Bayesian methods (lines 138-147) needs better context as it relates to the content of the research presented. One gets the impression that the author has a few issues with Bayesian methods but this complex topic needs either more or less detail when presented as a statistical practice that may be growing in popularity.

d. The John et al. paper is by no means the last word, in fact it was only the first word, in surveys on questionable conduct. Among other things, it failed to distinguish those who used the practice on one occasion from those who did it habitually. This meta-analysis of over a dozen similar studies is more up to date: Fox, N., Honeycutt, N., & Jussim, L. (2022). Better Understanding the Population Size and Stigmatization of Psychologists Using Questionable Research Practices. Meta-Psychology, 6.

e. The opinion on corrections for multiple testing is prone to misinterpretation without context. Corrections are only appropriate when all the tests in the family are testing the same null hypothesis of complete equivalence, e.g. among the means of a multilevel factor where one is only looking for any differences as a condition of H1. When testing formally separate hypotheses (including different tests in the same paper, separately theoretically meaningful main effects and interactions in an ANOVA) they are not needed.

Rubin, M. (2021). When to adjust alpha during multiple testing: A consideration of disjunction, conjunction, and individual testing. Synthese, 199(3), 10969-11000.

Gelman, A., Hill, J., & Yajima, M. (2012). Why we (usually) don't have to worry about multiple comparisons. Journal of research on educational effectiveness, 5(2), 189-211.

2. In the target journals, the author does not regularly acknowledge that there have often been concrete changes in policy in reporting, which is likely to have shaped the changes shown, depending on the enforcement by editors. By looking at author guidelines and editorials it would be possible and desirable to see how much of the change is journal-driven and how much is author-driven. It can also be seen whether journals are keeping authors to their own standards!

3. line 294, "many articles report overly many p-values" - by what standard? This seems more of a personal prejudice than a reasoned argument on desirable statistical content. Some journals publish nine-study papers, for example, and a total of 90 tests can easily come out of such a large program. Thus, the raw number of p-values does not seem to be a good metric of usage let alone a cause for alarm.

4. In table 2, the proportion of p-values in each significance band is of much more interest than the raw figures, and I would want to see them tested pre- and post-2015 like the other parameters in table 1. However, the analysis does not distinguish between p-values that attach to focal claims, and those that are from auxiliary tests, exploratory tests, tests of equivalence (for gender effects, for example, where gender is not the focus of the research), or manipulation checks. I am not necessarily asking for a more rigorous classification, I know it would depend on content and is not machine-readable. But -- it is important to avoid giving the misleading impression that journals are free to publish "nonsignificant findings" when these are mere footnotes to the main significant findings. In that case perhaps the general prevalence of p-values is not that important after all.

5. Are the changes before and after 2015 in Table 1 themselves significant (or Bayes different, etc.)? This is a recurring problem, claiming trends without considering that they could be statistical flukes.

6. Again regarding multiple corrections, it is not clear whether the kind of research in a field lends itself differentially to multiple corrections, but I think the author's misunderstanding of the need for this feature leads them to believe that for example every test in a 9-study paper should correct for every other test (alpha = .0001?) yet this might be different if split up into three three-study articles, which is clearly absurd. To be accurate, this treatment would need to go beyond the general vague sense that corrections should be applied, and look at the appropriateness - whether the family is indeed multiple tests that share a common null hypothesis of equivalence.

7. On sample size, the experiment design is a huge factor that is not discussed and can lead to misleading conclusions when comparing journals. Fields such as psychophysiology that use within-designs can achieve smaller sample sizes with no detriment to actual statistical power. However, for temporal comparisons we are probably on better grounds saying sample sizes have gone up or not in a meaningful way.

8. It would be interesting to see if one-tailed testing is especially prevalent for p-values that are two-tailed between .06 and .10. While it is correct to apply this method where hypotheses are stated a priori (e.g. pre-registered) and interest in the other direction of effects is formally renounced, my experience is that it is often used cynically and inappropriately to "rescue" a marginally significant result, even when the other tests are two-tailed.

9. line 436: " Although the power of a statistical test could simply be increased by one-sided tests, hardly any study uses directed tests" - increasing power is not a good reason to use directed tests; increasing power by changing the alpha criterion should only be done with a strong justification of having a laxer standard for confirmed results. Having a strong directional hypothesis and NO interest in surprising outcomes in the other direction is the main legitimate justification. If you find it goes the other way, d= 1.0, you still have to report a one-tailed test as nonsignificant!

10. The Discussion is mostly summative and does not reflect on the factors that could have contributed to the changes that appear to be true (and we are not sure how solid these changes are because, ironically, there are no p-values, confidence intervals or Bayesian tests associated with the claims of change). It is clear that there is a change in some areas but not every field and journal is buying in, while the strongest changes seem to be driven outright by journal policies. A theoretical model where change and field differences are explainable by a) researcher adoption of changing standards in practices b) utility of those practices to research in the specific field and c) concrete journal requirements informed by a and b as well as d) editors choosing to enforce these requirements, would seem to provide the required intellectual framework.

Signed, by policy, Roger Giner-Sorolla

6. PLOS authors have the option to publish the peer review history of their article (what does this mean?). If published, this will include your full peer review and any attached files.

Reviewer #1: No

Reviewer #2: No

Reviewer #3: **Yes: **Roger Giner-Sorolla

---

## [Author Response · Author response to Decision Letter 0]

25 Aug 2022

Reply to Reviewer 1

a) Low power = small samples = high uncertainty

Statistical power is undoubtedly as positive property of a statistically analyzed study. As the author mentioned, however, so is noisy measurements. Also, power is linked to sample size and expected effect sizes, and it should be estimated before running a study. Of all these important points, the authors gather data only on frequency of a priori power analysis and sample size.

It would be interesting to see whether study with a priori power analysis showed greater power than study without an a priori power analysis (given the observed effect sizes) or at least whether study with an a priori power analysis employed a larger sample as compared with studies that do not run a power analysis. This would help to really understand the consequences of adherence to “best practices”. We all assume that best practices are beneficial for the quality of research, but nobody brings evidence that they are. A study like this one is an opportunity to check the effects of these practices.

- Thanks for the comment. I agree with you that this correlation is interesting, but I think that this question needs its own manually curated study, since the application of power analyzes is quite heterogeneous (a priori/post hoc, targetted effect size, statistical method), which cannot be distinguished well here. 

Coherently, I was expecting the author to gather information about reporting standardized effect size indices. It would be interesting to show effect size increased over time, and if there is a negative correlation with the sample size, as the use of power analysis would imply.

- Thanks for the helpful comment. I have added an analysis regarding the reporting of standard effect measures pre/post 2016 (line 440).

No mention of the level of measurements reliability in the analyzed paper can be found in the results section, even thought the author mentions this property as an important quality of research.

- Thanks for the comment. I have added information about the accurary of the allgorithms in the method section (line 306). 

b) The undirected nil-null test scenario

The author claims that directed tests (one-sided tests) improve the quality of researching. Under some circumstances, such as in registered report, many would agree with this claim. However ,what happens in practice is that researchers willing to use one-sided tests are often accused by reviewers to capitalize on their chances to get something significant, because one sided tests are more powerful, and thus more likely to yield a significant result than two-sided tests. 

So, the lack of one-sided tests (5% before 2015 and 3% after) may simply reflect the more stringent adherence of reviewers to the so-called “best practices”, which usually entails asking for more stringent tests.

- Thanks for the comment. I agree with you that post hoc decisions for one-tailed testing increases the false positive rate and should be labeled bad practice. I pointed this out explicitly in the discussion (line 556). Still, I believe, that one-sided tests are severely underrepresented and that undirected testing should be replaced with equivalence or two one-sided tests against minimal effects of interest. A good overview of this type of testing can be found here: 

https://journals.sagepub.com/doi/10.1177/2515245918770963

The other claim in this section regards the pledge to report confidence intervals. Confidence intervals, when computed based on large sample theory, contain exactly the same information than the frequentist inferential tests they come from. They just portrait the information in a different format. Furthermore, they are even more misinterpreted that the corresponding inferential tests. Thus, reporting confidence intervals does not add or subtract anything to the quality of the research, and possibly introduce misinterpretation. Interpreting correctly the intervals, however, can be a positive addition to an article. If I recall well, Cumming’s “new statistics” was not about blindly reporting the confidence intervals, but it was about interpreting them and draw conclusions from their width. If one only uses them to check if zero is inside, a t-test/p-value would do the job just fine.

- Thanks for the comment. I specified that (line 153 + 528). 

Do the author have evidence that articles that report confidence intervals also interpret them correctly, or they use them in any way which improves the evaluation of the results? Just showing that confidence intervals are reported more often now than in the past, may simply reflect changes in journals policy, without any practical effect on research quality. 

- Unfortunately, no statement can be made in this regard as the interpretation of CIs is not focussed here.

c) Questionable research practices

This issue is mentioned in the introduction, but no data are reported to say something about it. This issue may be removed from the introduction, or some creative data may be found to check on its evolution over time. Number of registered reports may be analyzed, or comparisons between registered and not registered studies can be carried out on some key properties. Replications of the same experiment within the same papers will be also interesting, but I guess that this information is more difficult to extract from the data.

- Thanks for the comment. An analysis of the evolution of preregistered studies, has been added in a new result section (line 440). 

d) Multiple testing

The statement saying that “[..] the α level or p-values should always be adjusted/corrected for multiple tests” is too general and likely untrue. Assume I have a 3 balanced groups (A,B,C) experiment, and I registered that my theory will be supported if, and only if, A !=C and (A+C)/2!=B. If both comparisons are false, the probability of supporting my hypothesis is alpha^2, which is obviously less than alpha. If only one comparison is “true” (say the second), then my probability of supporting my hypothesis is alpha*(1-beta2), which is always less than alpha, since 0 Adjustment for multiple comparisons is required when one is willing to accept any statistically significant results as valid and supporting a hypothesis. 

- Thanks for the comment. I changed the statement according the adjustment of α level from „always“ to „usually“ (line 218). 

So, the author should check if adjustment is more likely when no hypothesis is laid out in advance, because that is when adjustment is requirement. Showing that adjustment has increased in frequency may just be a sign that researchers are reducing their a priori hypotheses and increased their HARKing.

- Thanks for the comment. Since no information is extractable on whether hypotheses were laid out in advance, this question remains unanswered.

2) Has the statistical analysis been performed appropriately and rigorously? NO

If I had the option, I would have marked this “partly”. The methods are good and fine, but I’d like to mention the necessity to add control variables to the analyses. The most striking example is the analysis of the number of p-values reported by each paper. Number of p-values may be highly correlated with number of studies reported in a paper and methods used (experimental vs correlational studies). 

- Thank you for this helpful comment. I now report the number of p-values per study instead of per article.

Thus, any increase or decrease over time of the number of reported p-values per article may be uncorrelated with the inferential strategies of the researchers, but simply due to a change in format of the journals. In non-randomized designs, furthermore, number of p-values can also be correlated with the use of control variables, as a way to identify possible spurious effects. It would be ironic if more controlled studies were judged as less accurate because they present more p-values than less controlled studies. 

- Thanks for the comment. I agree with you on this. 

As regards sample size (which is gathered for studying reporting it), it will make more sense to control for effect size. A small study on a large effect size may be as powerful as a large study on a small effect size. In general, sample size is correlated with the generality of the effects one is studying. Surveying detailed opinions about a product may require a much larger sample to obtain stable results than a study counting how many hemispheres there are in the human brain. As mentioned above, it would be also interesting to correlate the sample size with the presence of a priori power analysis.

- As stated above, such an analysis should be examined more closely in a separate study.

I would suggest to elaborate more the theoretical justification and the implications of the different indicators analyzed in the paper, and find more interesting relationships among indicators that could give a more precise and insightful overview of the evolution of the field.

Reply to Reviewer 2

1. In the introduction, the author fails to cite the literature militating against significance testing. The author should cite that literature or explain why it is not relevant.

- Thanks for the comment. The literature has been cited (line 164).

2. The issue of power analysis seems problematic too in that if one eschews significance testing, then why perform a power analysis to determine the sample size needed to have a good chance of getting a significant result?

- I do not mean to eschew significance testing, rather I am trying to promote justified analysis strategies with specific, non nil-null hypotheses and adequate sample sizes.

3. Why is there no discussion of the fact that, as most people use confidence intervals the way they use significance tests, there is no difference in practical usage? And why is there no discussion of the fact that a confidence interval is severely limited by the fact that the researcher has no idea of the probability that the population parameter is in the interval?

- Thanks for the comment. I have added a statement regarding this justified concern (line 548):

4. The author does not explain why the analyses performed benefit the field.

- Thanks for the comment. I have added a statement in the discussion (line 508).

Reply to Reviewer 3

1. The paper has a not very deep appreciation of some of the criticisms of psychological methods and their limits. At times the writing is overly long and at a very basic level for the issues that are studied in the meta-science research that follows.

We can assume that most researchers know the basics of hypothesis testing without needing to see the formula for a t-test, for example, and citations of sources about the controversy can be cited in passing. 

- Thank you for the comment. The formula has been removed. 

The Introduction needs to focus on the reason why these standards might be changing and the voices calling for change, rather than on basic-level explanations or unsupported claims. In other words, it needs to spend more time being descriptive about why change might be observed, and less being prescriptive and hinting at the author's opinions about the correct course of action. 

a. The Ioannidis claim that most published research findings are false is only valid under conditions of testing very low probability hypotheses, ignoring the many non-flashy studies that test well-grounded ideas. If the rate of hypothesis truth was the 10% or so required for this claim to be true, then why would selections of Registered Reports return confirmation rates of 45% or so? This suggests that hypotheses being tested in psychology are about 50% true with some attrition for type II error under high power.

Scheel, A. M., Schijen, M. R., & Lakens, D. (2021). An excess of positive results: Comparing the standard Psychology literature with Registered Reports. Advances in Methods and Practices in Psychological Science, 4(2), 25152459211007467.

see also

Wegener, D. T., Fabrigar, L. R., Pek, J., & Hoisington-Shaw, K. (2022). Evaluating research in personality and social psychology: Considerations of statistical power and concerns about false findings. Personality and Social Psychology Bulletin, 48(7), 1105-1117.

- Thanks for the helpful comment. This quote is indeed very pessimistic, but is intended to emphasize the problems in psychological research. I have added the statement, that this claim is exaggerated (line 25) and cited the results of Scheel et al. (line 211).

b. The discussion of power assumes that studies can be spoken of objectively as high or low powered when actually it depends critically on the effect size of interest. The assumption of a medium-sized effect as desirable and a small effect as optimum is nowhere justified.

- Thank you for the comment. The text has been revised. 

c. The discussion of issues with Bayesian methods (lines 138-147) needs better context as it relates to the content of the research presented. One gets the impression that the author has a few issues with Bayesian methods but this complex topic needs either more or less detail when presented as a statistical practice that may be growing in popularity.

- Thank you for the comment. The text has been revised (line 166+). 

d. The John et al. paper is by no means the last word, in fact it was only the first word, in surveys on questionable conduct. Among other things, it failed to distinguish those who used the practice on one occasion from those who did it habitually. This meta-analysis of over a dozen similar studies is more up to date: Fox, N., Honeycutt, N., & Jussim, L. (2022). Better Understanding the Population Size and Stigmatization of Psychologists Using Questionable Research Practices. Meta-Psychology, 6.

- Thanks for this helpful comment. I have revised the section and cited the proposed literature (line 201).

e. The opinion on corrections for multiple testing is prone to misinterpretation without context. Corrections are only appropriate when all the tests in the family are testing the same null hypothesis of complete equivalence, e.g. among the means of a multilevel factor where one is only looking for any differences as a condition of H1. When testing formally separate hypotheses (including different tests in the same paper, separately theoretically meaningful main effects and interactions in an ANOVA) they are not needed.

Rubin, M. (2021). When to adjust alpha during multiple testing: A consideration of disjunction, conjunction, and individual testing. Synthese, 199(3), 10969-11000.

Gelman, A., Hill, J., & Yajima, M. (2012). Why we (usually) don't have to worry about multiple comparisons. Journal of research on educational effectiveness, 5(2), 189-211.

- Thanks for the helpful comment. I revised the section and discussion of results regarding corrections for multiple testing and cited the proposed literature.

2. In the target journals, the author does not regularly acknowledge that there have often been concrete changes in policy in reporting, which is likely to have shaped the changes shown, depending on the enforcement by editors. By looking at author guidelines and editorials it would be possible and desirable to see how much of the change is journal-driven and how much is author-driven. It can also be seen whether journals are keeping authors to their own standards!

- Thanks for the comment. I have specified this in the discussion (line 516). 

3. line 294, "many articles report overly many p-values" - by what standard? This seems more of a personal prejudice than a reasoned argument on desirable statistical content. Some journals publish nine-study papers, for example, and a total of 90 tests can easily come out of such a large program. Thus, the raw number of p-values does not seem to be a good metric of usage let alone a cause for alarm.

- Thank you for the helpful comment. I replaced the number of p-values per article by the number of p-values per study to improve the metric. 

4. In table 2, the proportion of p-values in each significance band is of much more interest than the raw figures, and I would want to see them tested pre- and post-2015 like the other parameters in table 1.

However, the analysis does not distinguish between p-values that attach to focal claims, and those that are from auxiliary tests, exploratory tests, tests of equivalence (for gender effects, for example, where gender is not the focus of the research), or manipulation checks. I am not necessarily asking for a more rigorous classification, I know it would depend on content and is not machine-readable. But -- it is important to avoid giving the misleading impression that journals are free to publish "nonsignificant findings" when these are mere footnotes to the main significant findings. In that case perhaps the general prevalence of p-values is not that important after all.

- Thanks for the comment. As Table 1 already indicates, there are only minor changes in the pre/post 2016 proprtion of p-values and recomputable p-values that fall in each significance band. A more detailed analysis for each journal and year was omitted.

5. Are the changes before and after 2015 in Table 1 themselves significant (or Bayes different, etc.)? This is a recurring problem, claiming trends without considering that they could be statistical flukes.

- I can follow your concern and added a paragraph in the method section explaining why no p-values nor CIs are reported (line 308). 

6. Again regarding multiple corrections, it is not clear whether the kind of research in a field lends itself differentially to multiple corrections, but I think the author's misunderstanding of the need for this feature leads them to believe that for example every test in a 9-study paper should correct for every other test (alpha = .0001?) yet this might be different if split up into three three-study articles, which is clearly absurd. To be accurate, this treatment would need to go beyond the general vague sense that corrections should be applied, and look at the appropriateness - whether the family is indeed multiple tests that share a common null hypothesis of equivalence.

- Thanks for the comment. I revised the section and discussion of results on corrections for multiple Testing.

7. On sample size, the experiment design is a huge factor that is not discussed and can lead to misleading conclusions when comparing journals. Fields such as psychophysiology that use within-designs can achieve smaller sample sizes with no detriment to actual statistical power. However, for temporal comparisons we are probably on better grounds saying sample sizes have gone up or not in a meaningful way.

- Thanks for the comment. I have added a remark on this to the discussion (line 556).

8. It would be interesting to see if one-tailed testing is especially prevalent for p-values that are two-tailed between .06 and .10. While it is correct to apply this method where hypotheses are stated a priori (e.g. pre-registered) and interest in the other direction of effects is formally renounced, my experience is that it is often used cynically and inappropriately to "rescue" a marginally significant result, even when the other tests are two-tailed.

- Thanks for the comment. As no direct association of a single test result and the use of one-sided testing can be made, this interesting pattern will remain unclear. Post hoc decisions for one-tailed testing are obviously a questionable research practice. To avoid this misinterpretation I added the requirement to always pre-register studies with one-sided tests in the discussion (line 569-574).

9. line 436: " Although the power of a statistical test could simply be increased by one-sided tests, hardly any study uses directed tests" - increasing power is not a good reason to use directed tests; increasing power by changing the alpha criterion should only be done with a strong justification of having a laxer standard for confirmed results. Having a strong directional hypothesis and NO interest in surprising outcomes in the other direction is the main legitimate justification. If you find it goes the other way, d= 1.0, you still have to report a one-tailed test as nonsignificant!

- Thanks for the comment. I have revised the discussion regarding one-sided testing (line 564-568).

10. The Discussion is mostly summative and does not reflect on the factors that could have contributed to the changes that appear to be true (and we are not sure how solid these changes are because, ironically, there are no p-values, confidence intervals or Bayesian tests associated with the claims of change). It is clear that there is a change in some areas but not every field and journal is buying in, while the strongest changes seem to be driven outright by journal policies. A theoretical model where change and field differences are explainable by a) researcher adoption of changing standards in practices b) utility of those practices to research in the specific field and c) concrete journal requirements informed by a and b as well as d) editors choosing to enforce these requirements, would seem to provide the required intellectual framework.

- Thanks for this comment. The discussion has been revised.

---

## [Decision Letter · Decision Letter 1]

15 Nov 2022

PONE-D-22-14556R1Changes in methodological study characteristics in psychology between 2010-2021PLOS ONE

Dear Dr. Böschen,

Thank you for submitting your manuscript to PLOS ONE. After careful consideration, we feel that it has merit but does not fully meet PLOS ONE’s publication criteria as it currently stands. Therefore, we invite you to submit a revised version of the manuscript that addresses the points raised during the review process.

Both reviewers provide a series of specific points that you should consider. Please address all of them carefully in your revision.

We look forward to receiving your revised manuscript.

Kind regards,

Enrico Toffalini, Ph.D

Academic Editor

PLOS ONE

Reviewers' comments:

Reviewer's Responses to Questions

**Comments to the Author**

1. If the authors have adequately addressed your comments raised in a previous round of review and you feel that this manuscript is now acceptable for publication, you may indicate that here to bypass the “Comments to the Author” section, enter your conflict of interest statement in the “Confidential to Editor” section, and submit your "Accept" recommendation.

Reviewer #1: (No Response)

Reviewer #4: (No Response)

2. Is the manuscript technically sound, and do the data support the conclusions?

Reviewer #1: No

Reviewer #4: Partly

3. Has the statistical analysis been performed appropriately and rigorously? 

Reviewer #1: No

Reviewer #4: N/A

4. Have the authors made all data underlying the findings in their manuscript fully available?

Reviewer #1: Yes

Reviewer #4: Yes

5. Is the manuscript presented in an intelligible fashion and written in standard English?

Reviewer #1: Yes

Reviewer #4: Yes

6. Review Comments to the Author

Reviewer #1: The revision of the manuscript “Changes in methodological study characteristics in psychology between 2010-2021” addressed some of the reviewers concerns, but I still envisage a wide range for improvement.

The fundamental issue with the manuscript that informed my previous review and the other reviewers comments (as far as I understood them) is the strong discrepancy between the aim of the article and the data that are presented. The article aim, even more so in the revised version, is prescriptive, whereas the data are only descriptive. In the introduction and in the discussion of the paper we found several prescriptive statements regarding what should be done as a good practice and what should be avoided. Let aside that the proposed good practices are still discussed with little context and implications, the main issue is that the data described in the paper do not tell us much about how much psychological scientists are following those practices, and even less about the quality of their statistical machinery. Data describe some overall trends in research parameters, such as sample size, statistical approach, reporting styles. Evaluation of adherence to the good practices and quality of research required a deeper and more sophisticated analysis, as proposed in the first round of reviews, but the authors deem these analyses not doable with the available data.

I am aware that several of the more sophisticated analyses proposed by the reviewers cannot be carried out with the available data, but this limitation cannot be simply dismissed as a technical one. This limitation suggests that the article aim should be changed. The aim cannot be judging the quality of the published research, or its adherence with good practices, but rather describing the overall trends in statistical practice. As it is now, the paper portraits the wrong message than research lines that changed some of the selected parameters over time, such as reporting less p-values or adjusting for multiple comparisons, are improving their quality, when in reality the evidence reported in the paper is indecisive on this matter.

This general issue is reflected in many specific issues in the paper, especially after revision. I highlight the ones arising after the revision:

*) The manuscript mention increasing sample size as a good practice, and this is one of the least debatable statement. The data show positive trends of this parameter over time. However, in the discussion is now written “The estimated sample sizes have increased within the examined period from a median of 105 in articles published in and before 2015 to 190 in articles published after 2015. Assuming equally large effect sizes of interest and research designs, this is an indication of increased test power. Since the experimental design also influences the power of a test, smaller sample sizes should not strictly be considered as an indication of low power. Compared to between-subject designs, within-subject designs, as often used in Psychophysiology, can achieve a high power with smaller samples.” This comment basically says that the observed increase of sample size is not an indicator of more powerful tests, and so is saying that research may have not improved in quality. But then, what does this parameter say in terms of replicability and best practices? What are the implications of this result?

*) One-tail testing is mentioned as a good practice. The authors state “Besides the

fact that any good theory implies the direction of an effect, directed testing represents a

simple and cost-saving way to increase the power of a statistical test. In order to avoid

doubts about retrospective decisions on test sidedness (which surely is a questionable

research practice), intentions for directed testing should always be pre-registered.” Why should direct testing be always pre-registered, more than two-tail tests? If one has a strong theory, the direction of the effect can be derived from the theory, so it cannot be changed post-hoc. Furthermore, testing a theory-based directional hypothesis with a two-tail test is equivalent to testing it with a one-tail test at alpha/2, so the test is more stringent. Thus, maybe theory-based hypotheses should be pre-registered, independently of the number of tails one is willing to consider. More importantly, what does the data about one-tail testing has to do with the prescription of pre-registration? How do the authors derive this prescription from the available data? If the authors found a large percentage of one-tail test, one could suspect an abuse which could call for pre-registration. The authors found the opposite (very little use), so it is not clear how the prescription arises from the description. Again, one can be in agreement with the prescriptions mentioned in the paper, but it is difficult to see their relationship with the empirical results.

*) P-value: “Overall, 68% of the recalculated p-values are significant at α = .05. In half of the articles, more than two-thirds of the extracted and three-quarters of the recalculated p-values are below .05. This finding highlights the demand for highly-powered informed null-hypothesis testing with δ not equal to 0, or equivalence tests that reject the presence of a smallest effect size of interest”. How do these finding demand for highly powered informed null-hypothesis testing? It is not clear why 68% of significant p-values is an issue that will be solved by testing against δ not equal to 0? How many of this 68% were pre-registered? How many were simple descriptive correlations or preliminary group comparisons not testing any particular hypothesis of interest?

A final note concerns the justification for the lack of inference on the reported parameters. The authors justify the lack of inference by saying “The results are not reported with p-values nor confidence intervals because the sample is a full, not a random sample of psychological research articles by 12 highly reputed journals”. This justification is questionable. The authors aim is to present changes in methodological study characteristics in psychology, and to do that they selected a sample of journals, within which they collected the articles. Thus, they have a sample of articles. Their population is the population of articles published in top journals of psychology, which is not limited to the journals examined in the paper. Otherwise, the articles would not be representative of psychological research, and therefore no longer interesting.

Reviewer #4: This is a nice demonstration of the potential of JATSdecoder. The paper can, however, be improved on numerous fronts.

While I agree the usefulness of WEIRD research is limited, I don’t see a clear connection between having WEIRD samples and the replicability crisis, as replication studies typically tend to suffer from the same WEIRD myopic approach. Can the author further elaborate on how focusing on WEIRD participants might have contributed to the low replication rate observed in OSC? (The link would be obvious if the failure to replicate WEIRD studies was observed in non-WEIRD studies, but I don’t think that is the case).

Similarly, while I am aware of the many criticisms on p-values, I would like to read how exactly the author thinks that the reliance on p-values in general, or on nil-null hypotheses in particular, is a potential cause of the replication crisis.

Besides pre-registration, another solution to deal with researcher degrees of freedom (discussed in the QRP section) involves multiverse analyses (Steegen et al., 2016).

I understand that a fuller description and evaluation of JATSdecoder can be found in another publication, but still, I would like to read a bit more detail in the current manuscript. For example, how is the use of Bayesian methods identified? Is the occurrence of the term “Bayesian” or “Bayes factor” enough? I guess something more clever had been used (because by this procedure, articles that just discuss Bayesian statistics, rather than apply it, or also included), but I would like to get a rough idea of how this was solved.

I am not sure that using the year 2015 (when the OSC report was published) is a valid or useful marker for possible changes in study characteristics. If the OSC was a catalyst for change, its effect will only be visible a couple of years after 2015, given that research and publication is a slow process. And even if, say, 2017 is used as a demarcation line, it will be hard to disentangle the influence of the OSC and the general increasing awareness about replicability issues (of which the OSC itself was an exponent) and changing journal policies. This is acknowledged in the discussion, but using the publication date of OSC as the demarcation line attributes too much causal weight to OSC (and has, per above, the wrong date, even if it would be warranted to attribute that much causal weight to the OSC). More generally, I doubt the usefulness of any pre-201X vs post-201X analyses, for any X. What can we glean from such a dichotomous analysis that we cannot learn from looking at the year-by-year trends?

Looking at country of origin is a limited proxy to the WEIRDness of the population pool, at best, especially given the rise of worker markets like mturk and prolific. It is perfectly possible (and even likely) that a researcher from a Western university using mturk as a recruitment tool will collect data from, say, Asian participants. Also the other way around is possible (though less likely). I understand this problem is not easy to solve at the scale the current paper looks at, but it might be an issue worth mentioning.

It is very hard (for me at least) to grasp trends from tables like, Table 5. Maybe this information is best presented using a figure (with the corresponding tabled in the appendix for the full quantitative details).

I was somewhat surprised to see that even if journal policies are in place, the recommended or required practices are taken up at a limited scale only. Maybe this observation deserves more discussion. If policies don't seem to work, what other incentives does the author see to accelerate uptake of statistical reforms?

typos:

line 245: extend

line 332: 17$

signed,

wolf vanpaemel

7. PLOS authors have the option to publish the peer review history of their article (what does this mean?). If published, this will include your full peer review and any attached files.

Reviewer #1: No

Reviewer #4: No

---

## [Author Response · Author response to Decision Letter 1]

22 Dec 2022

Reply to Reviewer 1

The revision of the manuscript “Changes in methodological study characteristics in psychology between 2010-2021” addressed some of the reviewers concerns, but I still envisage a wide range for improvement.

The fundamental issue with the manuscript that informed my previous review and the other reviewers comments (as far as I understood them) is the strong discrepancy between the aim of the article and the data that are presented. The article aim, even more so in the revised version, is prescriptive, whereas the data are only descriptive. In the introduction and in the discussion of the paper we found several prescriptive statements regarding what should be done as a good practice and what should be avoided. 

Let aside that the proposed good practices are still discussed with little context and implications, the main issue is that the data described in the paper do not tell us much about how much psychological scientists are following those practices, and even less about the quality of their statistical machinery. Data describe some overall trends in research parameters, such as sample size, statistical approach, reporting styles. Evaluation of adherence to the good practices and quality of research required a deeper and more sophisticated analysis, as proposed in the first round of reviews, but the authors deem these analyses not doable with the available data.

I am aware that several of the more sophisticated analyses proposed by the reviewers cannot be carried out with the available data, but this limitation cannot be simply dismissed as a technical one. This limitation suggests that the article aim should be changed. The aim cannot be judging the quality of the published research, or its adherence with good practices, but rather describing the overall trends in statistical practice. As it is now, the paper portraits the wrong message than research lines that changed some of the selected parameters over time, such as reporting less p-values or adjusting for multiple comparisons, are improving their quality, when in reality the evidence reported in the paper is indecisive on this matter.

-> Thank you for the comments. I have revised the article to your concern. However, in my article, I want to highlight and question the overly ritualized use of nil-null hypothesis tests as psychology appears to be facing a replication crisis with many false positives, and empirically support many other authors' calls to move away from this practice (undirected test with H0: theta=0 at alpha=.05), to conduct power analyses, report effect sizes and CIs. In addition, I would like to encourage the preregistration of at least confirmatory research plans.

This general issue is reflected in many specific issues in the paper, especially after revision. I highlight the ones arising after the revision:

*) The manuscript mention increasing sample size as a good practice, and this is one of the least debatable statement. The data show positive trends of this parameter over time. However, in the discussion is now written “The estimated sample sizes have increased within the examined period from a median of 105 in articles published in and before 2015 to 190 in articles published after 2015. Assuming equally large effect sizes of interest and research designs, this is an indication of increased test power. Since the experimental design also influences the power of a test, smaller sample sizes should not strictly be considered as an indication of low power. Compared to between-subject designs, within-subject designs, as often used in Psychophysiology, can achieve a high power with smaller samples.” This comment basically says that the observed increase of sample size is not an indicator of more powerful tests, and so is saying that research may have not improved in quality. But then, what does this parameter say in terms of replicability and best practices? What are the implications of this result?

-> Thanks for the comment. I have added an analysis of median sample sizes in articles with and without repeated measure designs (Table 14) and can now clerify on this: “Assuming that the effect sizes of interest have not changed, this is an indication of increased test power.“ (line 592)

*) One-tail testing is mentioned as a good practice. The authors state “Besides the fact that any good theory implies the direction of an effect, directed testing represents a simple and cost-saving way to increase the power of a statistical test. In order to avoid doubts about retrospective decisions on test sidedness (which surely is a questionable research practice), intentions for directed testing should always be pre-registered.” Why should direct testing be always pre-registered, more than two-tail tests? If one has a strong theory, the direction of the effect can be derived from the theory, so it cannot be changed post-hoc. Furthermore, testing a theory-based directional hypothesis with a two-tail test is equivalent to testing it with a one-tail test at alpha/2, so the test is more stringent. Thus, maybe theory-based hypotheses should be pre-registered, independently of the number of tails one is willing to consider. More importantly, what does the data about one-tail testing has to do with the prescription of pre-registration? How do the authors derive this prescription from the available data? If the authors found a large percentage of one-tail test, one could suspect an abuse which could call for pre-registration. The authors found the opposite (very little use), so it is not clear how the prescription arises from the description. Again, one can be in agreement with the prescriptions mentioned in the paper, but it is difficult to see their relationship with the empirical results.

-> Thanks for the comment. If it was standard to test one-tailed, there would be no need to preregister. But within the present two-tailed standard, one-tailed testing may be suspicious to readers, although it is adequate. I have clarified that (line 604). 

*) P-value: “Overall, 68% of the recalculated p-values are significant at α = .05. In half of the articles, more than two-thirds of the extracted and three-quarters of the recalculated p-values are below .05. This finding highlights the demand for highly-powered informed null-hypothesis testing with δ not equal to 0, or equivalence tests that reject the presence of a smallest effect size of interest”. How do these finding demand for highly powered informed null-hypothesis testing? It is not clear why 68% of significant p-values is an issue that will be solved by testing against δ not equal to 0? How many of this 68% were pre-registered? How many were simple descriptive correlations or preliminary group comparisons not testing any particular hypothesis of interest?

-> Thanks for the comment. I have revised the statement (line 551)

A final note concerns the justification for the lack of inference on the reported parameters. The authors justify the lack of inference by saying “The results are not reported with p-values nor confidence intervals because the sample is a full, not a random sample of psychological research articles by 12 highly reputed journals”. This justification is questionable. The authors aim is to present changes in methodological study characteristics in psychology, and to do that they selected a sample of journals, within which they collected the articles. Thus, they have a sample of articles. Their population is the population of articles published in top journals of psychology, which is not limited to the journals examined in the paper. Otherwise, the articles would not be representative of psychological research, and therefore no longer interesting.

-> Thank you for the comment. I can see your point and added 99.9% CIs for the global comparisons in Table 1 and new Table 14, which shows the median estimated sample sizes in articles with and without repeated measures.

Reply to Reviewer 4

While I agree the usefulness of WEIRD research is limited, I don’t see a clear connection between having WEIRD samples and the replicability crisis, as replication studies typically tend to suffer from the same WEIRD myopic approach. Can the author further elaborate on how focusing on WEIRD participants might have contributed to the low replication rate observed in OSC? (The link would be obvious if the failure to replicate WEIRD studies was observed in non-WEIRD studies, but I don’t think that is the case).

-> Thanks for the comment. I don‘t see a direct connection of the WEIRDness of samples and the low replication rates in the OSC report either. To me, this is just another critical point in psychological research practice, when the aim is to generalize about humans. I have clarified on that (line 240).

Similarly, while I am aware of the many criticisms on p-values, I would like to read how exactly the author thinks that the reliance on p-values in general, or on nil-null hypotheses in particular, is a potential cause of the replication crisis.

-> Thanks for the comment. I don‘t believe that the reliance on p-values is a critical factor as „Valid P-Values Behave Exactly as They Should“ (Greenland, 2019).Rather, I think that the pressure to publish ‘significant’ results in a setting without thoughts about power encourages the use of questionable research practices, the effect of which is greatest when testing point null hypotheses.

Besides pre-registration, another solution to deal with researcher degrees of freedom (discussed in the QRP section) involves multiverse analyses (Steegen et al., 2016).

-> Thanks for the comment. I have added a brief introduction to multiverse analysis (line 218) and the result of a search task on mentions of multiverse analyses (line 475). They are very rarely applied in practice. 

I understand that a fuller description and evaluation of JATSdecoder can be found in another publication, but still, I would like to read a bit more detail in the current manuscript. For example, how is the use of Bayesian methods identified? Is the occurrence of the term “Bayesian” or “Bayes factor” enough? I guess something more clever had been used (because by this procedure, articles that just discuss Bayesian statistics, rather than apply it, or also included), but I would like to get a rough idea of how this was solved.

-> Thanks for the comment. I have added a brief explanation of how the statistical methods are extracted (line 267). I try not to include discussion papers into the analysis by the preselection procedure of empirical research articles. 

I am not sure that using the year 2015 (when the OSC report was published) is a valid or useful marker for possible changes in study characteristics. If the OSC was a catalyst for change, its effect will only be visible a couple of years after 2015, given that research and publication is a slow process. And even if, say, 2017 is used as a demarcation line, it will be hard to disentangle the influence of the OSC and the general increasing awareness about replicability issues (of which the OSC itself was an exponent) and changing journal policies. This is acknowledged in the discussion, but using the publication date of OSC as the demarcation line attributes too much causal weight to OSC (and has, per above, the wrong date, even if it would be warranted to attribute that much causal weight to the OSC). More generally, I doubt the usefulness of any pre-201X vs post-201X analyses, for any X. What can we glean from such a dichotomous analysis that we cannot learn from looking at the year-by-year trends?

-> Thanks for the comment. Indeed, the year 2015 is an arbitrary marker. Still, it enables a more global summary on changes over time. I have revised the statement on splitting at the year 2015 and have removed the reference to the OSC report. (line 253 + 342)

Looking at country of origin is a limited proxy to the WEIRDness of the population pool, at best, especially given the rise of worker markets like mturk and prolific. It is perfectly possible (and even likely) that a researcher from a Western university using mturk as a recruitment tool will collect data from, say, Asian participants. Also the other way around is possible (though less likely). I understand this problem is not easy to solve at the scale the current paper looks at, but it might be an issue worth mentioning.

-> Thanks for the comment. I have revised the section on sample characteristics in the discussion (line 592 + 614) but avoided to cite any company. 

It is very hard (for me at least) to grasp trends from tables like, Table 5. Maybe this information is best presented using a figure (with the corresponding tabled in the appendix for the full quantitative details).

-> Thanks for the comment. Graphics that display the result tables are also very messy (too many journals and years and too big differences between journals) and also difficult to read. I prefer to present the tables with exact numbers.

I was somewhat surprised to see that even if journal policies are in place, the recommended or required practices are taken up at a limited scale only. Maybe this observation deserves more discussion. If policies don't seem to work, what other incentives does the author see to accelerate uptake of statistical reforms?

-> Thanks for the comment. I really had to think about this and didn't come up with any good predictions. Apart from the requirement to register the research with a-priori quality criteria other than theta=0 and to justifiy the sample size with considerations of uncertainty, all the ideas I had were too speculative and mostly unrealistic.

typos:

line 245: extend

line 332: 17$

- Thanks for the advice on typos, which have been corrected.

---

## [Decision Letter · Decision Letter 2]

23 Feb 2023

PONE-D-22-14556R2Changes in methodological study characteristics in psychology between 2010-2021PLOS ONE

Dear Dr. Böschen,

Thank you for submitting your manuscript to PLOS ONE. After careful consideration, we feel that it has merit but does not fully meet PLOS ONE’s publication criteria as it currently stands. Therefore, we invite you to submit a revised version of the manuscript that addresses the points raised during the review process.

Please see the remaining minor points by Reviewer 4, and especially the correction suggested in their second paragraph.

We look forward to receiving your revised manuscript.

Kind regards,

Enrico Toffalini, Ph.D

Academic Editor

PLOS ONE

Journal Requirements:

Additional Editor Comments (if provided):

Please see the remaining minor points by Reviewer 4, and especially the correction suggested in their second paragraph.

Reviewers' comments:

Reviewer's Responses to Questions

**Comments to the Author**

1. If the authors have adequately addressed your comments raised in a previous round of review and you feel that this manuscript is now acceptable for publication, you may indicate that here to bypass the “Comments to the Author” section, enter your conflict of interest statement in the “Confidential to Editor” section, and submit your "Accept" recommendation.

Reviewer #4: (No Response)

2. Is the manuscript technically sound, and do the data support the conclusions?

Reviewer #4: Yes

3. Has the statistical analysis been performed appropriately and rigorously? 

Reviewer #4: Yes

4. Have the authors made all data underlying the findings in their manuscript fully available?

Reviewer #4: No

5. Is the manuscript presented in an intelligible fashion and written in standard English?

Reviewer #4: Yes

6. Review Comments to the Author

Reviewer #4: Ultimately, it is up to the author to decide on the framing of his work, but I still find the framing in terms of the replicability crisis confusing. The author acknowledges himself that some of the methodological aspects studies have little to no bearing the said crisis, so there is a possibility for agreement here. My preferred approach would be to discuss the methodological aspects without much fuzz, and in the discussion make the connection to the crisis, for those aspects for which such a connection is valid.

Am I correct to assume that a sentence like "The overall rate of articles using Bayesian inferential methods" is false, and should be "The overall rate of articles mentioning Bayesian inferential methods"? I understand non-empirical papers have been filtered out, but still, an empirical paper saying in the discussion that a fruitful future direction would involve Bayesian analyses would be categorized as using Bayesian analyses.

7. PLOS authors have the option to publish the peer review history of their article (what does this mean?). If published, this will include your full peer review and any attached files.

Reviewer #4: **Yes: **wolf vanpaemel

---

## [Author Response · Author response to Decision Letter 2]

24 Feb 2023

Reply to Reviewer 4

Ultimately, it is up to the author to decide on the framing of his work, but I still find the framing in terms of the replicability crisis confusing. The author acknowledges himself that some of the methodological aspects studies have little to no bearing the said crisis, so there is a possibility for agreement here. My preferred approach would be to discuss the methodological aspects without much fuzz, and in the discussion make the connection to the crisis, for those aspects for which such a connection is valid.

- Thank you for the comment. I prefer to stay with the current structure of the article and have not made any changes in this respect. 

Am I correct to assume that a sentence like "The overall rate of articles using Bayesian inferential methods" is false, and should be "The overall rate of articles mentioning Bayesian inferential methods"? I understand non-empirical papers have been filtered out, but still, an empirical paper saying in the discussion that a fruitful future direction would involve Bayesian analyses would be categorized as using Bayesian analyses.

- Thanks for the comment. As only the identified method and result sections are screened to extract the methodological study features, discussion sections containing references to Bayesian analyses should not lead to detections of Bayesian analyses. Nevertheless, I followed your suggestion and toned down the statements in line 447 + 449.

---

## [Editor Report · Decision Letter 3]

7 Mar 2023

Changes in methodological study characteristics in psychology between 2010-2021

PONE-D-22-14556R3

Dear Dr. Böschen,

We’re pleased to inform you that your manuscript has been judged scientifically suitable for publication and will be formally accepted for publication once it meets all outstanding technical requirements.

Kind regards,

Enrico Toffalini, Ph.D

Academic Editor

PLOS ONE
---

## [Editor Report · Acceptance letter]

13 Mar 2023

PONE-D-22-14556R3 

Changes in methodological study characteristics in psychology between 2010-2021 

Dear Dr. Böschen:

I'm pleased to inform you that your manuscript has been deemed suitable for publication in PLOS ONE. Congratulations! Your manuscript is now with our production department. 

Kind regards, 

on behalf of

Dr. Enrico Toffalini 

Academic Editor

PLOS ONE